# Railway fastener defect detection using RFD-DETR: A lightweight real-time transformer-based approach

**Huixiang Zhou, Yuhao Liu**[ID]*, **Jian Wang**

School of Information and Software Engineering, East China JiaoTong University, Nanchang, Jiangxi, China

* sinclair335648@gmail.com

**Data availability statement:** All data underlying the findings of this study are publicly available from the Roboflow platform at: https://universe. roboflow.com/objectdetectiondeeplearning/ railwaytrack_fastener_defcts1. The dataset

## Abstract

Railway fasteners play a crucial role in ensuring track stability and safety; however, manual inspection is both inefficient and prone to errors. Deep learning models, such as YOLO, are widely utilized for defect detection. However, these methods demand comparatively larger model sizes, greater computational power, and more storage capacity, yet their defect feature extraction remains inadequate. To overcome these challenges, this research introduces RFD-DETR, an enhanced detection transformer model specifically optimized for real-time rail fastener defect identification. The model incorporates three distinct modules to facilitate multi-scale feature extraction, enhance model efficiency, and improve defect detection. Firstly, a wavelet transform convolution module (WTConv) is employed, which integrates a wavelet transform to enhance multi-scale feature extraction while reducing model computation. Secondly, a cross-scale feature fusion module (CSP-PDC) is utilised, incorporating channel gated attention downsampling (CGAD) to refine defect detection. Finally, a wavelet transform feature upgrading (WFU) module is integrated within the neck module, enhancing feature fusion and contributing to the overall efficacy of the model. Experimental findings based on an expanded rail fastener dataset indicate that RFD-DETR achieves a 98.27% mean average precision (mAP) when evaluated at an IoU threshold of 0.5, outperforming the baseline model. Furthermore, it lowers computational expenses by 18.8% and reduces parameter count by 14.7%.

## Introduction

Rail fasteners are critical components of railway infrastructure, serving essential functions such as securing rails, absorbing train-induced vibrations, and maintaining track stability. However, railway fasteners are easily affected by environmental factors, train loads and human factors during long-term service, which may result in loosening, fracture, displacement, missing and other defects. These defects will not only reduce the stability of the track system, but also may cause serious railway safety accidents. Therefore, real-time and accurate detection of railway fasteners has important engineering value and safety significance. The traditional method of detecting rail fasteners mainly relies on manual inspection, i.e.

**Funding:** Huixiang Zhou, National Natural Science Foundation of China (Grant No. 62162028).

**Competing interests:** The authors have declared that no competing interests exist.

the maintenance personnel walk along the track and rely on visual observation of the fastener status. When anomalies are found, manual records and arrange for follow-up maintenance. Although this method has a certain degree of intuition, but its limitations are very obvious: first, manual inspection has low efficiency, making it challenging to fulfill the real-time monitoring demands of large-scale railway networks; secondly, human factors can introduce significant variability in detection results, as subjective judgments are highly susceptible to bias; thirdly, prolonged manual inspection involves intensive labor and poses significant safety risks. The rapid expansion of the rail network makes the traditional manual inspection method unable to adapt to the higher demand for efficiency and accuracy of modern track maintenance.

Track fastener detection using traditional image processing methods (e.g., edge detection and morphological operations) is used to isolate the fastener region, and then state recognition is achieved by template matching or feature analysis [1,2]. In contrast, the recent advances in computer vision technology have paved the way for innovative detection solutions for track fasteners [3,4]. Hong Fan et al. [5] proposed a fastener detection method for high-speed rail systems by integrating the Minima Significant Region (MSR) technique with Center-Symmetric Local Binary Patterns (CSLBP). This approach improves detection performance through precise fastener localization and low-dimensional feature extraction. However, it faces limitations in adaptability to complex environments and insufficient capability in detecting partially missing fasteners. R. Manikandan et al. [6] developed an automated machine vision-based inspection system that uses a Support Vector Machine (SVM) classifier to achieve intelligent identification and classification of missing fasteners in railway track images. Additionally, due to the complex track environment and changing fastener states, such methods are prone to fail under light changes, stain interference, and changes in fastener morphology, making it difficult to achieve high robustness and generalisation capability. Many traditional detection methods require the execution of complex image processing processes, and this multi-step processing mechanism not only introduces redundant computations, but also seriously affects the real-time detection performance of the system.

The rapid progress of deep learning technology has promoted the wide application of convolutional neural network (CNN)-based detection methods in the field of railway fastener detection. The current mainstream CNN detection methods mainly present two technical routes: the single-stage detection algorithm represented by the YOLO series, and the two-stage detection algorithm represented by Faster R-CNN. The latter, which rely on a two-step process—initially generating candidate boxes followed by refined object recognition—include methods such as R-CNN [7], Fast-R-CNN [8], Faster R-CNN [9], and Mask R-CNN [10]. In this study, an innovative two-stage detection framework [11] is proposed to significantly improve the accuracy of fastener detection by integrating the improved Fast R-CNN with the Support Vector Data Description (SVDD) algorithm. However, this architecturally complex model suffers from a large number of parameters and high computational cost, which makes it difficult to meet the dual demands of real-time and lightweight in railway fastener detection scenarios. In contrast, the single-stage detection methods represented by the SSD [12] and YOLO series [13–16] use end-to-end prediction to directly output the object location and category information, eliminating the time-consuming candidate region generation step in the traditional two-stage methods, and showing obvious advantages in detection efficiency. Notably, the YOLO series of algorithms has become an important research direction in the detection of railway fastener defects due to its end-to-end detection architecture and excellent real-time performance. Relevant researches include: the improved YOLOv5s model proposed by Li et al [17], which is improved by introducing CBAM attention module, BiFPN structure and K-means++ anchor frame optimization algorithm; MYOLOv3-Tiny network proposed by

Qi et al. [18] which is optimized by using depth-separable convolution and linear bottleneck inverted residuals structure; YOLO- developed by Chu et al. [19] O2E variant, which integrates a multi-branch parallel Enhanced Field of View module (EFOV), orthogonal dynamic convolution, and (OD_MP) multi-scale Attention module (EMA). These enhancements markedly improve both the precision and stability of the detection process, all while ensuring that speed remains uncompromised. Zhang et al. [20] introduced a lightweight track fastener detection approach that utilizes the YOLOv8n framework. Their method integrates an optimized EIOU loss function with a knowledge distillation strategy, thereby boosting detection accuracy and accelerating inference. Recently, Chen et al. [21] proposed DP-YOLO, which introduces depthwise separable convolution and attention mechanisms to achieve lightweight and accurate detection of rail fastener defects. Zhu et al. [22] proposed an improved YOLO11 model incorporating SPD-Conv for low-resolution feature enhancement and EffectiveSE attention for channel-wise feature refinement, achieving state-of-the-art performance in rail track defect detection.

However, the YOLO series algorithms still face challenges in detecting complex fastener defects. Initially, such methods need to first generate a large number of candidate regions (region proposals), and then apply a Non-Maximum Suppression (NMS) algorithm to filter these candidate frames in order to eliminate redundant and low-quality detection results. This process inevitably leads to a significant increase in computational demands. In contrast, Transformer-based methods offer a more lightweight and efficient approach. With its robust self-attention mechanism, Transformer model can accurately classify fastener defects in complex environments. Nevertheless, this high-performance detection mechanism is accompanied by significant computational overhead, making Transformer-based detection algorithms suffer from an efficiency disadvantage compared to other deep learning methods in scenarios with high real-time requirements. In the research of Transformer-based detection frameworks, representative models such as DETR [23], Deformable DETR [24], DINO [25], LW-DETR [26] and Co-DETR [27] have demonstrated better performance than the traditional YOLO family of algorithms in terms of detection accuracy by introducing an innovative attention mechanism. However, their substantial computational requirements hinder their performance in real-time detection scenarios. In 2023, Zhao et al. [28] developed RT-DETR, which is the first end-to-end Transformer architecture to achieve real-time detection, and its innovative use of intra-scale feature interaction and cross-scale feature fusion strategies effectively improves the characterization of multi-scale features. Lv et al. [29] proposed RT-DETRv2, which improves flexibility and deployment compatibility by introducing a "bag-of-freebies" strategy. This includes setting different sampling point numbers for multi-scale features in the deformable attention module and replacing the deployment-constrained grid_sample operation with a discrete sampling operator. Optimized training strategies are also adopted to enhance performance without increasing computational cost. Wang et al. [30] proposed RT-DETRv3, the latest version of the RT-DETR series, which aims to address the issue of insufficient training caused by sparse supervision signals in the DETR series. The model improves training efficiency and detection accuracy through hierarchical dense positive sample supervision. It introduces auxiliary branches with multiple target queries and a self-attention perturbation mechanism to enhance feature learning capabilities. By providing richer supervisory signals during the training phase, the model converges more quickly and achieves improved detection performance. To optimize the performance of small object detection, Zhang et al. [31] proposes the TSD-DETR model, which achieves 96.8% mAP on the standard test set by constructing a multi-level feature pyramid and designing a lightweight multi-scale attention mechanism, which is an improvement of 2.5 percentage points over RT-DETR. Li et al. [32] introduced IF-DETR for insulator defect detection, which specifically

addresses small defect challenges in complex environments through multi-scale feature fusion and a specialized IDIoU loss. Their approach achieved a 7.47% average precision improvement over existing methods, demonstrating the critical importance of preserving small target spatial information. While their work focuses on electrical insulators, it validates our approach of enhancing multi-scale feature representation for small infrastructure component defects. Aiming at the special needs of fastener defect detection, Song et al. [33] improved RT-DETR, and the main contributions include: developing a super-resolution convolution module (SRConv) to enhance local detail features; and incorporating channel attention in the self-attention mechanism. Experimental validation shows that these improvements significantly enhance the detection accuracy while maintaining computational efficiency. In addition, to address special scenarios in railway fastener detection, such as occlusion and complex background interference, Bai et al. [34] proposed a detection method based on TSR-Net. This method integrates visual Transformers, inverted residual blocks, and self-supervised transformation attention mechanisms to enhance the model's detection capability in complex scenes. Experimental results indicate that this method achieves high accuracy and robustness when detecting fasteners occluded by foreign objects. However, its computational cost is relatively high, posing limitations for applications with strict real-time requirements.

To overcome the shortcomings of conventional approaches in terms of computational overhead and feature identification performance, this research introduces an efficient real-time railway fastener inspection approach leveraging an enhanced RT-DETR architecture. To enhance detection performance, the RT-DETR model is optimized by incorporating lightweight convolutional modules during the feature extraction stage, reducing computational complexity and improving detection speed. Additionally, the self-attention mechanism is refined to focus more effectively on fastener defect regions while minimizing sensitivity to irrelevant background information. This study ensures efficient real-time detection while improving fastener defect detection accuracy, providing an effective and feasible intelligent detection solution for railway maintenance and safety monitoring. The main innovative contributions of this study are in the following areas:

1. Introducing the Wavelet Transform Convolution Module (WTConv) to fuse high- and low-frequency features, enhancing feature extraction while reducing computational complexity;
2. Improving the Cross-Scale Feature Fusion Module using Cross-Stage Partial Parallel Dilated Convolution (CSPPDC) and Channel Gated Attention Downsampling (CGAD) to enhance the effective features and suppress the redundant features;
3. Feature fusion optimization innovatively introduces the wavelet transform feature upgrading (WFU) module, which effectively fuses high-frequency detailed features with low-frequency structural information through the multi-scale decomposition and reconstruction mechanism.

## Methods

RT-DETR is an innovative detection framework which has been developed to achieve real-time detection performance. The framework is capable of remarkable performance due to its advanced design. This detection framework integrates three principal architectural elements: a feature extraction backbone, a multi-modal encoder module, and a decoder unit incorporating transformer mechanisms with auxiliary prediction capabilities. Benchmark analyses against established YOLOv8 demonstrate RT-DETR's enhanced computational efficiency and

optimized performance balance in equivalent experimental settings. Significantly, the architecture attains training acceleration while preserving detection velocities parallel to YOLO-series benchmarks, achieving these advancements without implementing mosaic-style data enrichment protocols.

The model is available in multiple versions, such as R18, R34, R50, R101 [35], L, and X, each corresponding to different levels of computational complexity and parameter scales. Considering the complex inspection environment and constraints on model size in rail fastener inspection, RT-DETR-R18, which has smaller computational volume and higher accuracy, is chosen as the baseline algorithm. Fig 1 systematically illustrates the architectural composition and operational workflow of this detection system.

Transformer-based detection architectures typically integrate three core operational modules: feature extraction backbone, encoding unit, and decoding mechanism. Initial processing involves convolutional operations (e.g., ResNet [35]) transforming input images into multi-dimensional feature representations. Spatial dimension adaptation is achieved through vectorization of these feature tensors, subsequently augmented with positional embeddings - critical components that maintain spatial awareness for effective geometric relationship interpretation within Transformer frameworks. These processed tensors then propagate through the encoder-decoder cascade, where the decoder component strategically incorporates trainable query vectors to generate predictive outputs. Final detection parameters, comprising coordinate regression coefficients and categorical probabilities, are computed through fully-connected layer transformations of these decoder outputs.

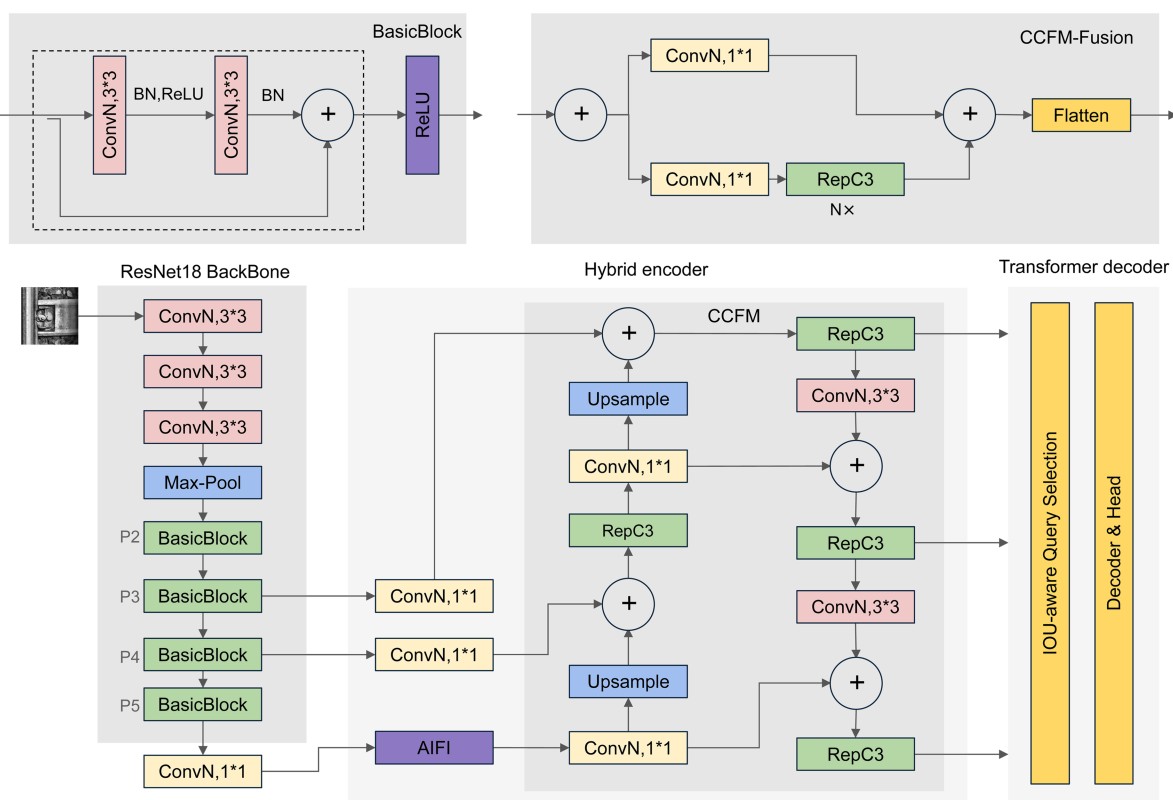

**Fig 1. Original RT-DETR model structure diagram.**

Notably, while maintaining architectural commonalities with conventional Transformer detectors, our implementation introduces substantial innovations in encoding processes. As depicted in Fig 1, the backbone generates hierarchical feature maps (P3-P5) through progressive downsampling, with P5 demonstrating superior semantic abstraction capabilities. Crucially, the system implements feature pyramid network (FPN) methodology to amalgamate P3 and P4 features into the P5 representation. This strategic fusion preserves multi-scale contextual information while concentrating encoding operations exclusively on the enhanced P5 layer, achieving computational redundancy reduction compared to conventional multi-layer encoding approaches.

## RFD-DETR

Accuracy is a fundamental goal and primary requirement for object detection tasks, especially in railway fastener detection. However, the real-time object detection transformer (RT-DETR) model has some limitations in detecting railway fastener defects, including insufficient feature extraction, low computational resource utilization, and redundant channel attention computation.

This paper proposes an enhanced RT-DETR model that overcomes the limitations of existing detection methods, as illustrated in Fig 2. First, the model's primary innovation is the effective integration of high-frequency and low-frequency features, facilitated by the incorporation of a wavelet transform convolution module (WTConv). The fusion of these methods

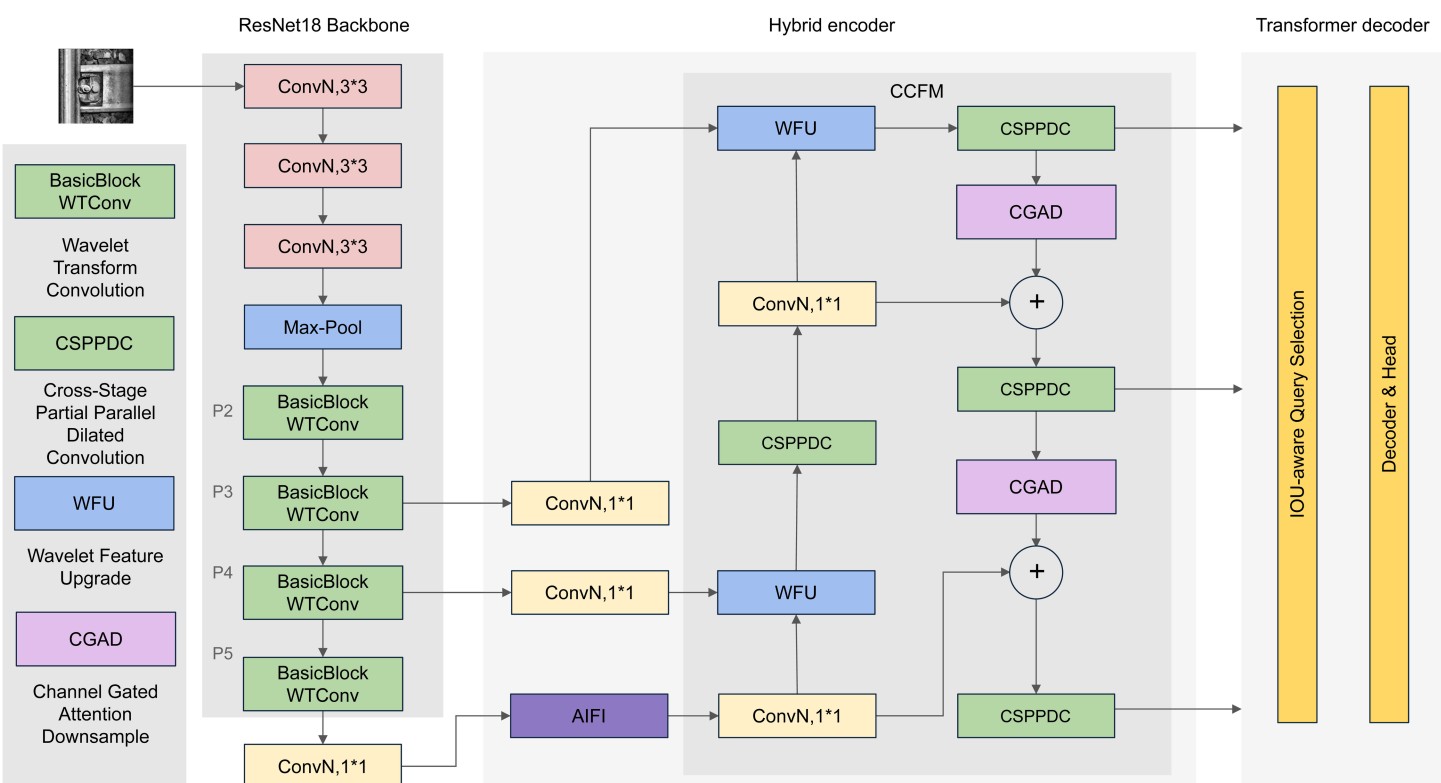

**Fig 2. RFD-DETR network structure diagram.** The diagram illustrates the overall architecture of the RFD-DETR model, including the backbone, neck module, and detection head. Key modules such as WTConv, CSPPDC, and WFU are highlighted, showing the flow of feature extraction, fusion, and enhancement.

enables efficient feature extraction at multiple scales while reducing computational requirements and memory consumption. The resultant model demonstrates enhanced capacity to detect diverse objects in complex scenes, thus demonstrating significant advancements in the field. Furthermore, an independent design module named PDC-CGAD is proposed. It leverages the cross-stage partial network (CSPNet) [36] with parallel dilated convolution (CSP-PDC) at varying atrous rates to efficiently capture local and multi-scale features. Additionally, the Channel Gated Attention Downsampling (CGAD) mechanism is incorporated to amplify effective features while suppressing redundant ones, thereby optimizing the original model's performance. Lastly, a Wavelet transform feature upgrading (WFU) module [37] is introduced to enhance feature fusion and detail preservation. The WFU module decomposes the input features into low-frequency and high-frequency components through Haar wavelet transform, processes these two components independently respectively, and finally re-fuses the processed features so as to effectively retain the structural information and texture details of the image. This integrated approach significantly reduces the amount of computation and number of parameters while improving the detection accuracy, ultimately constructing an efficient and lightweight model.

## Backbone network integrated with wavelet transform convolution module

We propose the WTConv module, which innovatively integrates wavelet transform into the convolutional backbone. This enables efficient multi-frequency feature extraction and significantly reduces computational complexity, making it the first time wavelet-based convolution is applied to lightweight real-time railway defect detection.

In order to improve the efficiency of railway track fastener detection and reduce computational redundancy, this paper selects the lightweight ResNet-18 as the basic backbone network. Furthermore, improvements are made to ResNet-18, the backbone of the RT-DETR network, by introducing a convolution module (WTConv [38]) that integrates wavelet transform to replace the standard convolution operation. This enhancement forms WTConv-Blocks, which significantly boost feature extraction capability and improve multi-scale target recognition performance. The WTConv2d module realizes efficient extraction of multi-scale frequency domain features through multi-level wavelet decomposition and reconstruction. Consequently, the proposed adaptation not only strengthens the model's capacity to capture features at multi-scale frequencies but also markedly reduces computational demands and memory consumption.

Specifically, the BasicBlock in ResNet-18 consists of two convolutional layers, both originally designed as standard 2D convolutional operations. To effectively capture multi-frequency features using wavelet transform, we replace the second convolutional layer of the BasicBlock with a custom WTConv2d module. This module decomposes the input feature map into different frequency bands (low frequency LL and high frequency LH, HL, HH) by wavelet transform and performs depthwise convolution and scaling operations for each sub-band respectively. Subsequently, the inverse wavelet transform reconstructs each sub-band into the original resolution feature map, which is then summed with the output of residual concatenation to complete multi-scale processing of the input features.

The WTConv2d architecture implements multiresolution analysis through spectral decomposition operators, executing discrete wavelet transformations that bifurcate input signals into distinct frequency sub-bands. This dual-channel processing mechanism operates as:

$$\mathbf{X}_{WT} = WT(\mathbf{X}) = Conv(\mathbf{X}, \mathbf{F}_{WT}) \tag{1}$$

where $X$ denotes input feature tensor, $F_{WT}$ is the wavelet filter, and $X_{WT}$ is the feature map after wavelet transformation, containing quadrature components: approximation coefficients (LL) capturing low-frequency patterns and detailed coefficients (LH,HL,HH) preserving high-frequency features. The inverse wavelet transform is used to reconstruct the feature map from the wavelet domain back to the spatial domain:

$$\mathbf{X}_{\text{IWT}} = \text{IWT}(\mathbf{X}_{\text{WT}}) = \text{Conv}^T(\mathbf{X}_{\text{WT}}, \mathbf{F}_{\text{IWT}}) \tag{2}$$

where $F_{IWT}$ is the inverse wavelet filter. WTConv2d achieves multi-scale feature extraction through cascaded wavelet decomposition. At each layer of decomposition, the input feature map is decomposed into low and high frequency components, each of which is independently subjected to depth-separable convolution and scaling. This process can be expressed as:

$$\begin{aligned}\mathbf{X}_{\text{WT}}^{(i)} &= \text{WT}(\mathbf{X}_{\text{LL}}^{(i-1)}) \\ \mathbf{Y}^{(i)} &= \text{Conv}(\mathbf{X}_{\text{WT}}^{(i)}, \mathbf{W}^{(i)})\end{aligned} \tag{3}$$

where $X_{WT}^{(i)}$ is the feature map after wavelet decomposition at level $i$, and $W^{(i)}$ is the convolutional kernel at level $i$. The convolved feature maps are then reconstructed level by level using the inverse wavelet transform:

$$\mathbf{X}_{\text{LL}}^{(i-1)} = \text{IWT}(\mathbf{Y}^{(i)}) \tag{4}$$

The final output is obtained as:

$$\mathbf{Y} = \text{IWT}(\mathbf{Y}^{(1)}) \tag{5}$$

WTConv2d achieves exponential growth in receptive field while maintaining linear growth in the number of parameters. For a k × k receptive field, the number of trainable parameters in WTConv2d scales logarithmically with k. This allows WTConv2d to achieve a larger receptive field compared to traditional convolutions while maintaining efficient parameter utilization.

## Improvement of cross-scale feature fusion module

We propose a novel Cross-Scale Feature Fusion Module that combines Cross-Stage Partial Parallel Dilated Convolution (CSPPDC) and Channel Gated Attention Downsampling (CGAD). This design enables efficient multi-scale context aggregation and selective feature enhancement, significantly improving the model's ability to capture both local details and global structures in complex railway defect scenarios.

In order to enhance performance in object detection tasks, it is vital to optimise the efficiency of feature fusion and expand the receptive field. As demonstrated in the preceding section, WTConv effectively expands the convolutional receptive field while exhibiting efficient utilisation of model parameters. The original RT-DETR model featured the RepC3 fusion block within the Convolution-based Features Fusion Module(CCFM). This block combined features from both branches by means of an element-wise summation. However, this direct fusion approach may fail to fully leverage the unique contributions of each branch, potentially leading to information loss. Additionally, the structure's limited feature representation capacity may hinder its ability to accurately extract and recognize complex objects. To

overcome these challenges, a novel cross-stage partially parallel dilated convolution (CSP-PDC) module, complemented by a channel-gated attention downsampling (CGAD) mechanism, has been proposed with the objective of enhancing the feature representation and expanding the receptive field.

The proposed CSPPDC architecture, as depicted in Fig 3, implements a multi-branch dilated convolution paradigm, where three parallel 3×3 convolutional operators with progressively increasing dilation factors (d=1, 2, 3) are strategically combined to extract hierarchical contextual features. This configuration effectively establishes an anisotropic receptive field expansion mechanism, enabling simultaneous perception of both local details and global structural patterns. The outputs from these convolutions are concatenated along the channel axis and subsequently refined through a 1×1 convolution to effectively merge the extracted features. To enhance cross-stage feature fusion and optimize input feature channel dimensions, the module incorporates two independent 1×1 convolution layers that adjust feature dimensions prior to further processing. The processed PDC features are then integrated with the dimension-adjusted input features along the channel space, followed by an additional 1×1 convolution to generate a refined feature map. By leveraging multiple dilation rates, the CSP-PDC module effectively enlarges the receptive field, enabling the model to capture spatial details across various resolutions. This design significantly enhances feature expressiveness and improves localization precision, particularly when detecting complex objects.

To further enhance the characterization of complex target features, the channel-gated attention downsampling (CGAD) method is introduced, as illustrated in Fig 3. The CGAD method leverages global average pooling (GAP) to extract comprehensive contextual information from the feature maps while employing a channel gating mechanism to emphasize critical features and suppress redundant ones. Specifically, the input feature map is first processed using GAP to capture its overall contextual representation. Subsequently, a 1×1 convolution is used to generate a channel attention map, and a Hardsigmoid activation function is

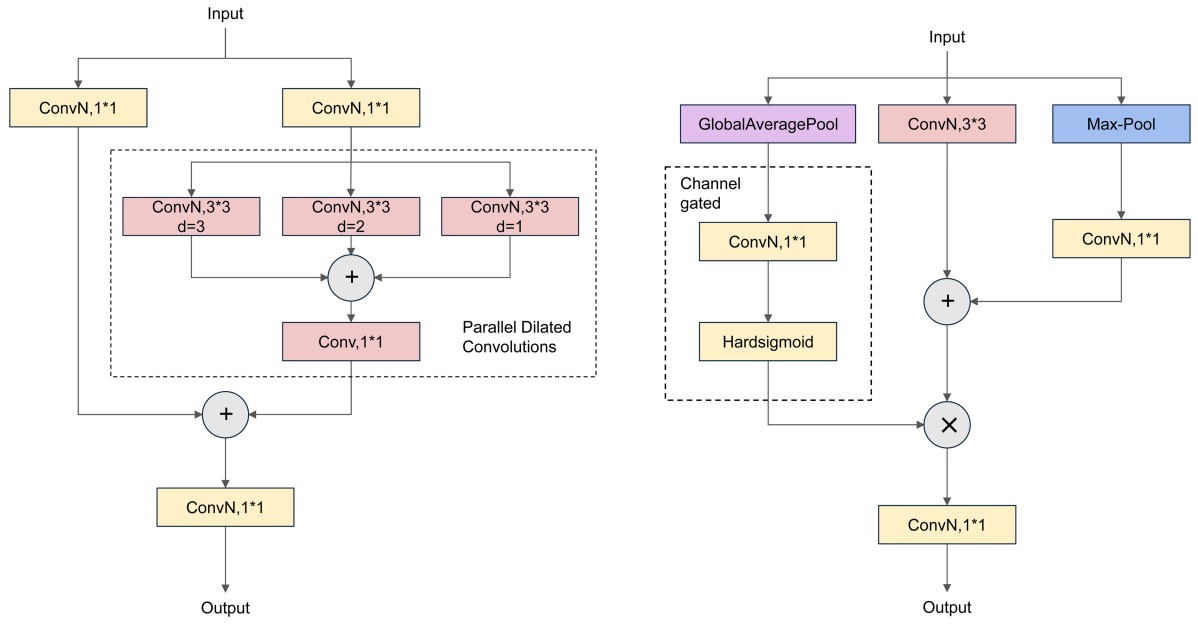

**Fig 3. Structure Diagram of PDC-CGAD**.

introduced to dynamically learn the importance weights of each channel. The original feature map is multiplied with the attention weights channel by channel to realize the enhancement of critical features and the suppression of non-critical features. Furthermore, downsampling is performed using either a 3×3 convolution or max pooling to reduce spatial resolution while preserving essential structural details. The outputs from these downsampling operations are fused along the channel dimensions to generate richer feature representations. This approach not only improves the diversity of extracted features but also enhances the representation of significant features while minimizing extraneous information.

By integrating the CSPPDC module with the CGAD method, the proposed model efficiently captures multi-scale contextual information while optimizing feature representations for complex targets, resulting in enhanced detection accuracy and computational efficiency. The synergy between multi-scale feature extraction and the channel-gating mechanism significantly improves the robustness and precision of the railway fastener object detection model, thereby making it more effective for real-world applications.

## Wavelet-transform feature upgrade of neck module

We propose the WFU (Wavelet transform feature upgrading) module, which innovatively applies wavelet decomposition and reconstruction in the neck of the network. This enables effective fusion of high-frequency detail and low-frequency structure, significantly enhancing feature representation and detail preservation for small and subtle defects in railway fastener detection.

The baseline hybrid encoder architecture integrates Attention Intra-Feature Interaction (AIFI) techniques by transforming the multi-scale feature map P5 into a serialized image representation. This design choice stems from P5's superior semantic abstraction capability, which compensates for information loss occurring during prior downsampling operations. Conventional feature fusion approaches often exhibit limited cross-scale integration capacity, particularly in preserving both high-frequency spatial details and low-frequency structural patterns during inter-scale propagation. In order to enhance the fusion effect of high-level features extracted by the AIFI module and to strengthen the detail performance of the image, this paper designs a feature enhancement structure based on wavelet transform, called WFU module.

The WFU module employs wavelet transformation to break down larger-scale features into low-frequency and high-frequency components, thereby enabling feature fusion across different scales. Fig 4 illustrates the initial step, where the encoder-derived feature map $F_s \in \mathbb{R}^{\frac{H}{4} \times \frac{W}{4} \times 4C}$ undergoes decomposition via the Haar wavelet transform. This transform, recognized for its simplicity and effectiveness in multi-resolution analysis, separates the features into approximation and detail elements. Upon applying the Haar wavelet transform, the feature map is decomposed into four distinct sub-bands: $A_{LL}$, $A_{LH}$, $A_{HL}$, and $A_{HH}$. Here, $A_{LL}$ captures the low-frequency components, encapsulating the image's general structure and the majority of its energy. In contrast, the high-frequency sub-bands—$A_{LH}$, $A_{HL}$, and $A_{HH}$—represent the image's finer details, corresponding to horizontal, vertical, and diagonal orientations, respectively. This decomposition facilitates a comprehensive representation, effectively preserving both the overarching structure and intricate textures of the image.

To enhance fastener details, the high-frequency components are processed independently. The three high-frequency sub-bands $A_{LH}$, $A_{HL}$, and $A_{HH}$ are summed to obtain a combined high-frequency feature representation. Subsequently, a residual block is applied to further enhance these features, effectively emphasizing image detail information while mitigating the gradient vanishing problem and improving feature representation capabilities.

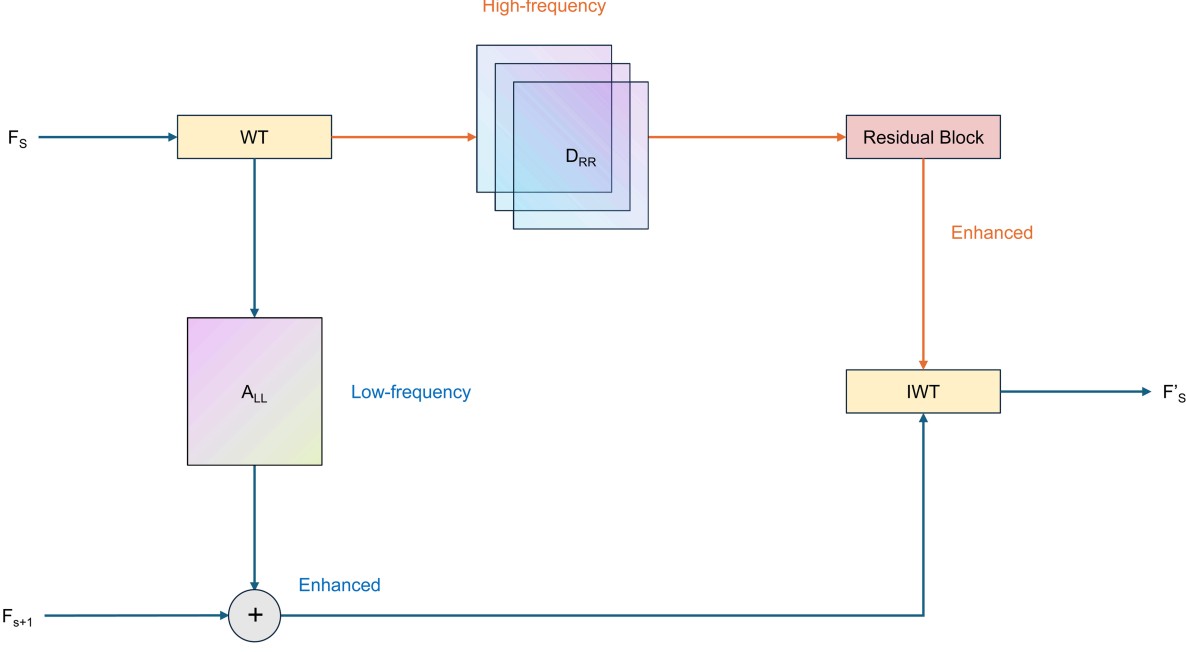

**Fig 4. Structure Diagram of WFU.**

Simultaneously, the low-frequency sub-band $A_{LL}$ is fused with the smaller-scale feature $F_{s+1}$ from the decoder. To achieve this, $F_{s+1}$ and $A_{LL}$ are combined along the channel dimension and passed through a channel transformation module. This module efficiently integrates features from varying scales, generating more expressive and representative feature representations. It comprises two convolutional layers responsible for dimensionality reduction and expansion, thereby enabling effective interaction between multi-scale features.

In order to obtain the final up-sampled features, the enhanced high-frequency components need to be integrated with the fused low-frequency components, and the image should be reconstructed by Inverse Haar Wavelet Transform (IWT). The inverse Haar wavelet transform recombines the decomposed components into complete image features, achieving both feature upsampling and detail enhancement.

By employing this approach, the WFU module effectively fuses multi-scale features and enhances image detail information. Unlike traditional upsampling methods, WFU module mitigates the aliasing problem caused by directly combining high- and low-frequency components, while the multi-resolution characteristics of wavelet transform better preserve the structure and texture information of fasteners.

## Experiment and results

### Dataset description

The principal dataset employed in this study is a railway track fastener detection dataset sourced from Roboflow [39], a comprehensive end-to-end computer vision platform. The dataset was provided in an augmented form, with image enhancement techniques such as flips, scaling, rotations, skeletonization, and other adjustment strategies already applied,

resulting in a total of 2,234 images. The original, unaugmented images are no longer available, and the specific augmentation parameters (e.g., the probability and range of flips, scaling, rotations, etc.) are not documented in the dataset description. Consequently, all training and evaluation in this study were conducted using the provided augmented dataset, and the precise details of the augmentation process cannot be fully reported. The dataset contains six categories of fastener states: e-type fastener, w-type fastener, e-type fastener-broken, w-type fastener-broken, fastener-missing, and fastener-foreign-object. This dataset is referred to as RFD in this paper. As shown in Fig 5, the figure presents the detailed tag categories information of the RFD dataset.

The dataset was partitioned into training and validation subsets at a 9:1 ratio. To evaluate generalization capability, a separate set of 124 images sourced from the platform was employed for model performance assessment.

## Evaluation indicators

In order to comprehensively evaluate the performance of the proposed algorithm, we conducted comparative experiments and ablation experiments using the following key metrics: precision (P), GFLOPs (giga floating-point operations per second), parameters (the total learnable variables in the model), and mean average precision (mAP)—a composite metric integrating classification accuracy and recall across categories.

$$P = \frac{TP}{TP + FP} \tag{6}$$

$$AP = \int_0^1 P(R)\,dR$$
$$mAP = \frac{\sum_{n=0}^{C} AP(c)}{C} \tag{7}$$

Where $P$ and $R$ denote precision and recall, $P(R)$ represents the precision-recall curve function, and $C$ corresponds to the total number of classes.

$$\text{GFLOPs} = W \times H \times K \times K \times C_{in} \times C_{out} \tag{8}$$

$$\text{Parameters} = C_{in} \times C_{out} \times K \times K \tag{9}$$

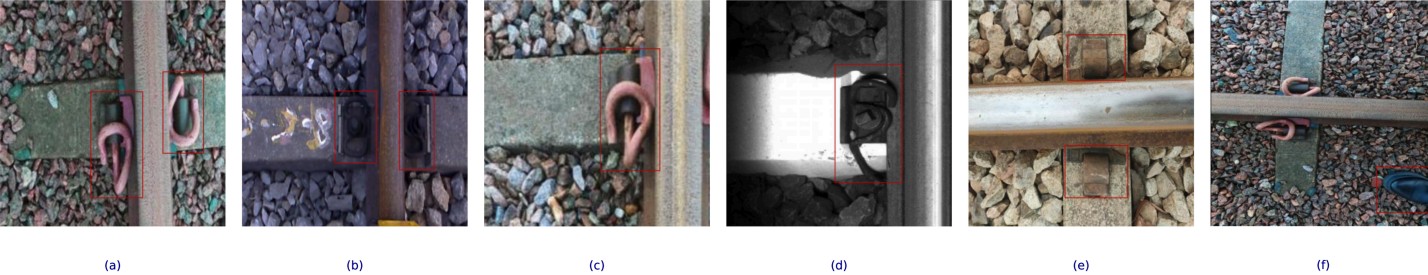

(a)   (b)   (c)   (d)   (e)   (f)

**Fig 5. Dataset sample.** Example images for each of the six fastener categories in the RFD dataset: (**a**) E-type fastener; (**b**) W-type fastener; (**c**) E-type fastener-broken; (**d**) W-type fastener-broken; (**e**) Fastener-missing; (**f**) Fastener-foreign-object. Each image shows the typical appearance and labeling of the corresponding category.

Computational complexity was assessed via GFLOPs and parameters, with $W$ and $H$ indicating feature map dimensions, $K$ the convolution kernel size, and $C_{in}$ and $C_{out}$ the input and output channel counts, respectively.

## Experiment environment and parameter setting

All experiments were conducted on Ubuntu 22.04 operating system using Python 3.10 and PyTorch 2.1.2. During training, the batch size was set to 16, the number of training rounds was 100, the initial learning rate was $1 \times 10^{-4}$, the IoU was 0.7, and the input image size was uniformly adjusted to $640 \times 640$ pixels. The specific hardware configuration and model parameters are detailed in Table 1.

## Comprehensive analysis of the RFD-DETR model

Table 2 shows the detection performance of RFD-DETR model for different categories of rail fasteners and anomalies. From the overall results, the model achieves high detection accuracy and recall in all categories, and the overall performance is excellent. The model achieved a mean average precision (mAP@50) of 98.27% across all categories, demonstrating high detection accuracy at an intersection-over-union threshold of 0.5. The mAP@50:95 declined to 72.39% with elevated IoU thresholds, suggesting scope for enhancing the model's bounding box regression precision under stricter localization criteria. In terms of specific categories, the recalls of W-type fastener, W-type fastener-broken and E-type fastener-broken all reach 100%, and the mAP@50:95 are all between 73.06% and 80.74%, indicating that the model is able to more stably detect different types of fasteners and their damage. In contrast, the mAP@50:95 for the Missing category is relatively low at 60.46%, which may be influenced by the complexity of the samples or the model's ability to learn features for this category. In addition, the Trackbed-stuff category has a recall of only 69.41% although the precision reaches 100.00%, indicating that there is a high leakage rate in this category, which may be due to the fact that its features are not sufficiently distinctive or there is an imbalance in the data distribution. Overall, the improved RT-DETR model performs well in the rail fastener detection task, especially in the detection of fasteners and their damage, but there is still room for further optimisation of the generalisation ability in some categories (e.g. Missing and Trackbed-stuff),

**Table 1. Experiment configuration and model parameters.**

| Hardware | Specification | Hyperparameter | Value |
|---|---|---|---|
| GPU | RTX 3090 | Learning rate | $1 \times 10^{-4}$ |
| CPU | 14 vCPU | Momentum | 0.9 |
| CUDA | 12.1 | Optimizer | AdamW |
| CuDNN | 8.9.0 | Batch size | 16 |

**Table 2. Performance metrics for different classes.**

| Class | Instances | Precision (%) | Recall (%) | mAP@50 (%) | mAP@50:95 (%) |
|---|---|---|---|---|---|
| All | 325 | 99.10 | 94.50 | 98.27 | 72.39 |
| E-type | 162 | 99.19 | 98.77 | 99.47 | 77.64 |
| W-type | 30 | 98.99 | 100.00 | 99.50 | 80.74 |
| W-type-broken | 21 | 97.99 | 100.00 | 99.50 | 77.63 |
| E-type-broken | 13 | 98.61 | 100.00 | 99.50 | 73.06 |
| Missing | 83 | 99.80 | 98.80 | 98.51 | 60.46 |
| Trackbed-stuff | 16 | 100.00 | 69.41 | 93.11 | 60.03 |

which can be further improved by data enhancement, loss function adjustment and feature extraction optimisation to improve detection precision and recall.

## Verification of generalizability for complex scenarios

In order to further analyze the robustness of the validation model in extreme weather and complex backgrounds, the performance of the model under different weather conditions and weather severity is compared. Table 3 compares the initial model with the improved model and conducts gradient tests for each weather type. Using image processing to enhance the data, fog, raindrops and snowflake effects are added to the validation set individually, and the degree of severity is light, medium and heavy, so there are a total of nine validation sets, and the table compares the performance of these nine validation sets, and the bolding in the graphs indicates which of the two models performs better under the same metrics comparison.

In the original scenario, the improved model (Ours) outperforms the baseline (RT-DETR-R18) in both the mAP50 and mAP50:95 metrics, which is used to validate the change in performance of the two models in extreme weather. The next analysis shows that under foggy conditions, the performance of the two models is close for light fog concentrations, with the improved model being slightly better; the performance of the models decreases for moderate fog concentrations, with the improved model being significantly better than the baseline model on mAP50 but slightly lower on mAP50:95, indicating that the model is stronger at detecting targets but slightly worse at localization accuracy; the performance under heavy fog concentrations shows a substantial degradation, but the improved model is significantly ahead of the baseline model in both metrics, showing stronger robustness, indicating that the improved model is still better at detecting and localizing targets under extreme foggy days. Mild The baseline model slightly outperforms the improved model in light rain, but the difference is small; the improved model outperforms the baseline model in medium rain, showing better adaptability; the baseline model outperforms the improved model in heavy rain in both indicators. Rain In general, the improved model has slightly better robustness in light and moderate rain, which is comparable to the baseline model, but slightly worse than the baseline in heavy rain, and the rainy weather does not have much effect on both models, and the actual adaptability is better. Snowy days Under snowy conditions, both indicators of the improved model are better than the baseline in all aspects, and the advantage expands with the increase of snow amount, indicating that the improved model is extremely resistant to high-frequency noise and has excellent robustness in snowy days.

Table 3. **Performance comparison under different weather conditions and augmentation levels.**

| Weather Condition | Level of Augment | Baseline (mAP50) | Ours (mAP50) | Baseline (mAP50:95) | Ours (mAP50:95) |
|---|---|---|---|---|---|
| origin | - | 96.48 | **98.27** | 71.86 | **72.39** |
| Fog | Light | **96.48** | 96.36 | 69.17 | **69.27** |
| | Medium | 91.52 | **93.07** | **65.68** | 63.96 |
| | Heavy | 61.71 | **73.18** | 39.98 | **44.61** |
| Rain | Light | **97.09** | 96.36 | **70.77** | 69.13 |
| | Medium | 95.75 | **96.79** | 69.15 | **69.72** |
| | Heavy | **96.20** | 94.22 | **70.21** | 67.49 |
| Snow | Light | 96.31 | **97.96** | 70.54 | **70.97** |
| | Medium | 96.56 | **97.49** | 70.41 | **71.37** |
| | Heavy | 94.46 | **96.50** | 67.96 | **69.50** |

Even though these comparisons are mainly for extreme weather, these weather enhancements inherently affect the phenomena of blurriness, partial occlusion, and background complexity in the image. Therefore, the model's enhanced robustness in these scenarios also indirectly indicates that the model is more adaptable to complex scenes.

## Comparisons with different detection models

To validate the effectiveness of RFD-DETR, we conducted experiments on the training set and validation set, selecting multiple models from the current mainstream Yolo series and DETR series for detailed comparison. These models are similar in terms of parameter count and computational complexity. Additionally, to demonstrate the performance of the RFD dataset on models with high computational complexity and parameter counts, we also selected two models from the Co-DETR series with a ResNet50 backbone as large-scale model comparisons.

The experimental results of the accuracy comparison among different models are summarized in Table 4, where categories 1–6 correspond to the validation mAPIoU scores with a threshold of 50 for different dataset categories: E-type and W-type fasteners (intact, broken), missing fasteners, and foreign objects. The RFD-DETR model demonstrates a balance between performance and complexity compared to existing object detection baseline models on these metrics. Compared to YOLO-based real-time object detectors, the RFD-DETR model outperforms YOLOv5m, YOLOv8s, YOLOv10m, YOLOv11m, YOLOv12m by 0.83%, 4.93%, 2.62%, 0.67%, and 0.95%, respectively, but is slightly lower than the YOLOv8m model by 0.13%. When comparing real-time object detectors based on DETR, the RFD-DETR model outperforms Co-DINO-R50, Co-Deformable-DETR-R50, DINO-R50, LW-DETR-m, RT-DETR-R18, RT-DETR-R34, RT-DETRv2-R18, RT-DETRv2-R34, and RT-DETRv3-R18 by 1.83%, 5.07%, 1.49%, 2.29%, 1.79%, 0.64%, 1.59%, 0.86% and 0.79%, respectively. However, it is slightly lower than DEIM-DFine-M, and RF-DETR-B by 0.02%, and 0.37%, respectively.

The efficiency comparison experimental results of different models are summarized in Table 5. All models are tested under the same input image size and batch size, where the input

**Table 4. Accuracy comparison of different models on six categories.**

| Model | 1 | 2 | 3 | 4 | 5 | 6 | mAP@50(%) | mAP@50:95(%) |
|---|---|---|---|---|---|---|---|---|
| Yolov5m | 99.49 | 99.00 | 95.71 | 99.50 | 98.98 | 91.95 | 97.44 | 73.34 |
| Yolov8s | 94.13 | 95.01 | 84.76 | 98.42 | 95.60 | 92.12 | 93.34 | 70.46 |
| Yolov8m | 99.50 | 99.31 | 99.00 | 99.50 | 98.95 | 94.14 | **98.40** | **74.22** |
| Yolov10m [40] | 98.98 | 97.17 | 96.83 | 92.83 | 99.10 | 89.00 | 95.65 | 70.76 |
| Yolov11m [41] | 99.48 | 99.08 | 96.14 | 99.32 | 99.49 | 92.08 | 97.60 | 72.92 |
| Yolov12m [42] | 99.48 | 98.88 | 96.83 | 99.32 | 99.49 | 89.92 | 97.32 | 72.51 |
| Co-DINO-R50 [27] | 99.90 | 94.16 | 100.00 | 100.00 | 97.61 | 86.95 | 96.44 | 71.31 |
| Co-Deformable-DETR-R50 [27] | 98.37 | 92.86 | 97.81 | 98.44 | 96.18 | 75.83 | 93.20 | 69.00 |
| DINO-r50 [25] | 98.89 | 94.06 | 100.00 | 100.00 | 99.07 | 87.67 | 96.78 | 71.31 |
| LW-DETR-medium [26] | 99.14 | 94.10 | 100.00 | 99.55 | 98.08 | 85.00 | 95.98 | 67.63 |
| RT-DETR-R18 | 99.44 | 99.50 | 99.50 | 95.52 | 98.53 | 86.41 | 96.48 | 71.86 |
| RT-DETR-R34 | 99.48 | 99.50 | 99.50 | 99.50 | 99.50 | 88.31 | 97.63 | 71.80 |
| RT-DETRv2-R18 [29] | 99.53 | 94.64 | 99.80 | 99.72 | 97.07 | 89.32 | 96.68 | 72.14 |
| RT-DETRv2-R34 [29] | 99.73 | 93.96 | 100.00 | 100.00 | 98.09 | 92.69 | 97.41 | 71.84 |
| RT-DETRv3-R18 [30] | 99.83 | 95.60 | 99.80 | 99.81 | 99.73 | 90.14 | 97.48 | 72.28 |
| DEIM-DFine-M [43] | 99.70 | 95.38 | 99.86 | 99.21 | 99.93 | 95.61 | 98.29 | 72.41 |
| RF-DETR-B [44] | 99.34 | 99.55 | 99.55 | 99.61 | 97.83 | 96.00 | **98.64** | **72.57** |
| **RFD-DETR (ours)** | 99.47 | 99.50 | 99.50 | 99.50 | 98.51 | 93.11 | 98.27 | 72.39 |

**Table 5**. **Efficiency comparison of different models.**

| Model | GFLOPs(G) | Parameters(M) | FPS(bs=1) |
|---|---|---|---|
| Yolov5m | 64.0 | 25.0 | 184.8 |
| Yolov8s | 28.4 | 11.1 | **218.9** |
| Yolov8m | 78.7 | 25.8 | 161.2 |
| Yolov10m [40] | 58.9 | 15.3 | 152.9 |
| Yolov11m [41] | 67.7 | 20.0 | 145.5 |
| Yolov12m [42] | 67.1 | 20.1 | 112.7 |
| Co-DINO-R50 [27] | 333.9 | 64.2 | 9.8 |
| Co-Deformable-DETR-R50 [27] | 108.7 | 64.2 | 21.6 |
| DINO-r50 [25] | 35.9 | 47.6 | 16.8 |
| LW-DETR-medium [26] | 42.8 | 28.2 | 58.5 |
| RT-DETR-R18 | 57.0 | 19.9 | **63.3** |
| RT-DETR-R34 | 88.9 | 31.1 | 50.0 |
| RT-DETRv2-R18 [29] | 25.8 | 20.1 | 29.4 |
| RT-DETRv2-R34 [29] | 40.9 | 31.3 | 24.2 |
| RT-DETRv3-R18 [30] | 25.8 | 21.0 | 32.0 |
| DEIM-DFine-M [43] | 56.4 | 19.2 | 37.6 |
| RF-DETR-B [44] | 51.8 | 31.9 | 42.9 |
| **RFD-DETR (ours)** | 46.3 | **16.9** | 47.4 |

image size is 640*640 batch size is 1. Compared to YOLO variants, under conditions where the number of parameters is not significantly different, the RFD-DETR model has a computational load of only 46.3 GFlops, which is lower than that of most YOLO series models. Although the RFD-DETR model's mAP50 and mAP50:95 metrics are lower than those of the Yolov8m model in terms of accuracy comparison, the Yolov8m model has a computational load of 78.7 GFLOPS and 25.8 million parameters, which are significantly higher than those of the RFD-DETR model. This indicates that the RFD-DETR model has faster inference speed and performs well in terms of parameter count. Additionally, in terms of FPS performance metrics, which indicate the number of image frames a model can process, the RFD-DETR model still falls short compared to lightweight models like the YOLO series. However, when considering high accuracy, the RFD-DETR model achieves a good balance between speed and accuracy. When comparing computational complexity and parameter counts among DETR variants, the RFD-DETR model has a computational complexity that is lower than 10.1 GFlops and 5.5 GFlops compared to DEIM-DFine-M and RF-DETR-B, respectively, and a parameter count that is lower than 2.3M and 15M, respectively. Meanwhile, RT-DETRv2-R18 has a computational complexity that is 20.5 GFlops lower than RFD-DETR. Looking at the FPS metric, RFD-DETR is significantly faster than most DETR series object detection models. Although DEIM-DFine-M and RF-DETR-B have advantages over the RFD-DETR model in terms of accuracy, the RFD-DETR model outperforms them by 9.8 and 4.5 in terms of FPS inference speed, while also having lower parameter counts and computational complexity, thereby demonstrating its effectiveness and superiority in real-time object detection.

In summary, the RFD-DETR model balances high accuracy, low latency, and low resource consumption compared to DETR series models. Through structural innovations, efficient feature fusion, and improvements to lightweight attention mechanisms, it significantly enhances object detection capabilities while maintaining low model complexity and high inference speed.

## Main result

In order to validate the effectiveness of wavelet convolution module in rail fastener detection, ablation experiments are evaluated for different wavelet decomposition layers and different wavelet basis functions.

As shown in Table 6, comparing the performance of different wavelet basis functions at the same number of wavelet decomposition layers, this paper selected Haar, db1, and db2, which are three wavelet basis function types commonly used in image processing, and all of them show favorable multi-resolution characteristics in the target detection task. The experimental section used a 5×5 convolution kernel. Among these, db1 wavelet basis performs the best in terms of the mAP metrics, and haar and db2 are slightly lower, which shows that the wavelet basis functions of the selection affects the effect of feature decomposition, db1 can better adapt to the texture and edge features of the rail fastener defects, which improves the feature characterization ability and detection accuracy of the model, so we choose db1 as the wavelet basis function. As shown in Table 7, this paper compare the performance of different wavelet decomposition layers under the db1 wavelet basis function, and with the increase in the number of wavelet decomposition layers, the model in terms of the mAP metrics shows some changes, the model achieves the highest mAP in 1-level decomposition, and Precision and Recall also reach 98.01% and 96.28%, when the number of decomposition layers increases to 2-level and 3-level, although Recall slightly improves, the mAP slightly decreases, which indicates that too many decomposition layers may lead to part of the high-frequency information by This indicates that too many decomposition levels may lead to the loss of some high-frequency information or over-dispersion of features, so in order to balance the global and local information and enhance the ability of multi-scale feature expression, 1-level is chosen as the final wavelet decomposition level.

The Neck module, as a key part of the multi-scale feature fusion and upgrading of the RT-DETR model, plays an important role in the module's ability to capture texture edge feature information and optimize the feature characterization of complex targets. Therefore, an effective improvement strategy is proposed to combine the CSPPDC and CGAD modules to enhance the ability of the cross-scale feature fusion module to pay attention to target details and reduce the ability to pay attention to irrelevant background.

The enhancement of the model feature expression ability by the proposed improvement is visualized in Fig 6, which compares the feature map information of the model before and after the improvement in each key improvement layer under the same input. The first column represents the inference results of the model before and after the improvement on the same test image, and it can be observed that the model before the improvement has misdetection, and

**Table 6**. Performance comparison under different wavelet bases.

| Level | Wavelet-type | Precision (%) | Recall (%) | mAP@50 (%) | mAP@50:95 (%) |
|-------|--------------|---------------|------------|------------|---------------|
| 1-level | haar | 95.78 | 95.32 | 96.02 | 69.51 |
| 1-level | db1 | **98.01** | **96.28** | 97.32 | **71.73** |
| 1-level | db2 | 96.90 | 94.43 | **97.86** | 71.50 |

**Table 7**. Performance comparison under different wavelet decomposition levels.

| Level | Wavelet-type | Precision (%) | Recall (%) | mAP@50 (%) | mAP@50:95 (%) |
|-------|--------------|---------------|------------|------------|---------------|
| 1-level | db1 | **98.01** | **96.28** | 97.32 | **71.73** |
| 2-level | db1 | 96.96 | 96.10 | 96.62 | 69.47 |
| 3-level | db1 | 96.23 | 96.16 | **97.50** | 68.95 |

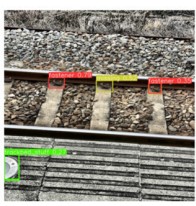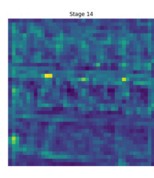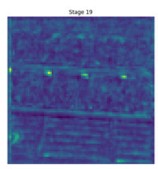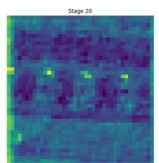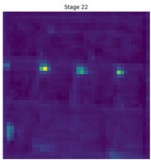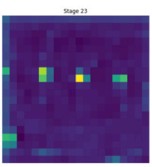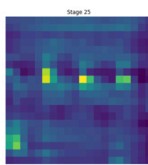

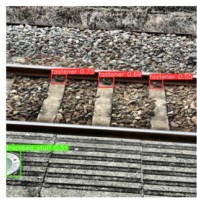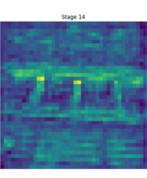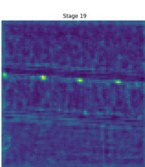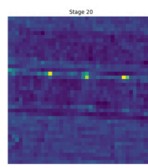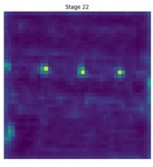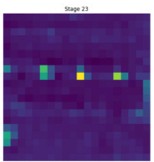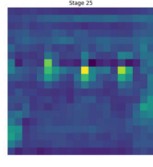

**Fig 6. Visualization of feature representation enhancement by CSPPDC and CGAD modules.** The figure compares feature maps before and after the introduction of CSPPDC and CGAD modules. Warmer colors indicate higher activation, highlighting improved focus on defect regions and suppression of background noise in the enhanced model.

the model after the improvement has more accurate detection frames in defective regions with higher confidence, reflecting the overall improvement of the model detection performance. Then later columns are the comparison of the feature maps after the model inference through each intermediate layer in turn, in the low-level region Stage 14 in the pre-improvement feature response is more dispersed, the defective region and the background of the distinction is not high, and the layer after the CSPPDC block after the improvement of the feature maps of the background region for effective suppression of the local structure and the edge of the information has a more pronounced expression of the information; in the middle and high-level regions, respectively, corresponds to 19, 20, 22, 23 and 25 layers, before improvement are RepC3 block, downsampling convolution, RepC3 block, downsampling convolution and RepC3 block, respectively, and after improvement are CSPPDC, CGAD, CSPPDC, CGAD, and CSPPDC, respectively, all of which are differentiated in the overall model structure after the addition of the PDC-CGAD module, and correspond to the pre improvement The response of the feature map to the defective region is gradually enhanced, and after the introduction of CGAD attentional downsampling can effectively improve the target differentiation ability, the feature map's attention to the defective region is further enhanced, and the activation of the non-target region is suppressed, and the feature focusing ability becomes stronger.

Overall, the addition of CSPPDC makes the low-level features more sensitive to the defective region, and the CGAD method makes the high-level features pay more attention to the target and the interference of the background is weakened through the channel-gated attention mechanism.

The RTDETR model enters the Neck module of the hybrid encoder, primarily responsible for feature fusion and processing. It enhances the interaction between features of different scales through the self-attention mechanism, particularly aiding in multi-frequency feature extraction in object detection, which helps the model better capture detailed information in images. Therefore, to make the model's representation of fastener defect detection more prominent in the feature space and effectively distinguish targets from backgrounds, the WFU wavelet feature enhancement is proposed to improve the model's ability to distinguish defects and enhance the detail information of image features.

Fig 7 shows the performance comparison of three interpretable heatmap methods when used on the same image. Each row represents the comparison results for the same original image, with the top row showing the original model and the bottom row showing the WFU-improved model. The first column displays the results after inference on the original image, while the remaining three columns show the results of the three activation mapping methods. Heatmaps visually illustrate the model's focus on target localization and distinction, primarily categorized into three color transformations: blue, green, and red. Red indicates the highest level of attention to the current region, green indicates the next highest level of attention, and blue indicates minimal attention to the region. In the Grad-CAM method, the improved model focuses more on the target area in the activation region; in the Grad-CAM++ method, both the original and improved models pay sufficient attention to the target, but the improved model has a larger red region, indicating that the model's basis for determining the target is more explicit; in the XGrad-CAM method, the original model has a larger activation region coverage area with scattered focus points, while the improved model has a uniform distribution, with the activation region highly overlapping with the defect location.

In summary, the WFU model significantly enhances the model's ability to express features in the target region. The improved model not only shows significant improvements in detection rate and accuracy but, more importantly, exhibits highly concentrated focus on the target

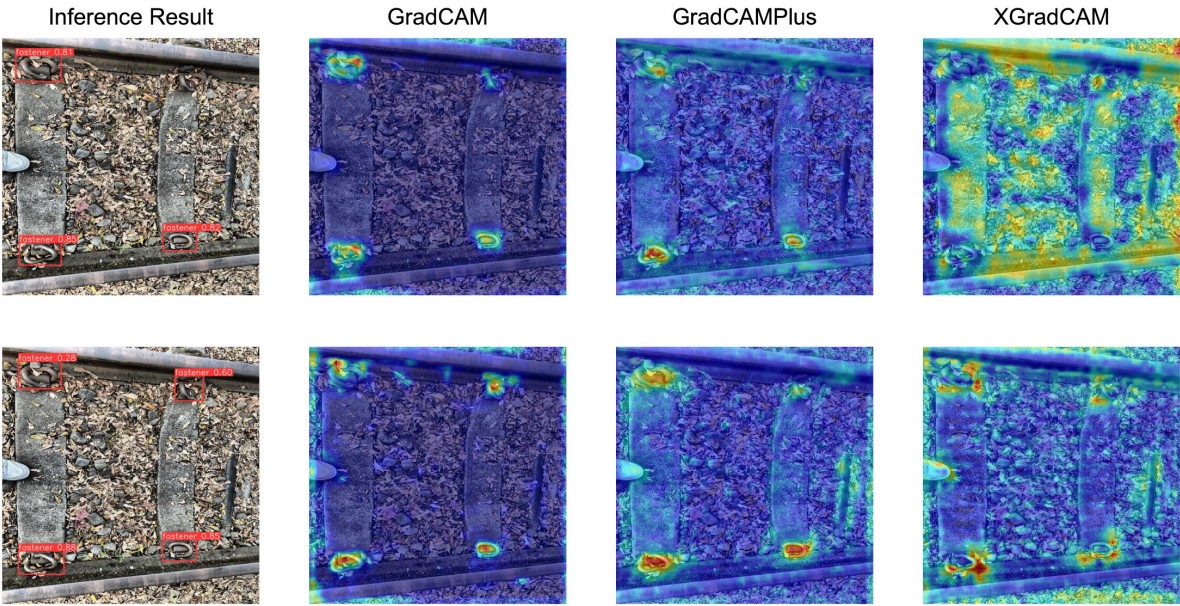

**Fig 7. Comparison of test results for different heat map methods.**

region, demonstrating the effectiveness and superiority of WFU in suppressing background interference and capturing feature information.

## Ablation study

To evaluate the proposed model's performance enhancements, eight ablation experiments were performed. Based on the RT-DETR network, several optimizations were made. The feature extraction network incorporated the lightweight WTConv module in place of the BasicBlock residual network's secondary 3×3 convolution block, optimizing feature extraction efficiency while minimizing computational demands. For cross-scale fusion, the CSPPDC module replaced both the RepC3 fusion block and downsampling mechanism, broadening the receptive field and strengthening multi-scale feature representation capabilities. Additionally, the downsampling process was refined using the CGAD method to enhance channel-wise attention and suppress redundant features. In neck module, some of the upsampling operations were removed to reduce computational overheads, and the concatenation block originally used to fuse large- and small-scale features was replaced with the WFU module, significantly improving feature fusion efficiency and preserving structural details.

Table 8 presents the experimental configurations and their impact on model performance using different combinations of methods, where a "✓" indicates the inclusion of the respective module. Ablation studies revealed that the incorporation of lightweight convolution (WTConv), a multi-scale feature enhancement module (PDC-CGAD), and wavelet feature upgrading (WFU) independently improved multiple performance metrics of the baseline. In Experiment 1, the application of WTConv alone results in a 0.84% increase in mAP50 (from 96.48% to 97.32%), alongside reductions of 35.5% in parameters (from 19.88M to 12.83M) and significantly lowering the computational cost to 40.3 GFLOPs, representing a 29.3% reduction. This outcome demonstrates that WTConv effectively mitigates model redundancy while boosting feature extraction efficiency. In Experiment 2, utilizing PDC-CGAD independently further elevates the mAP50 by 1.55% (reaching 98.03%). Despite a slight increase in model complexity (GFLOPs rising from 57.0 to 58.2), its capability to capture multi-scale features substantially enhances model accuracy. In Experiment 3, using WFU alone resulted in a slight decrease in mAP50 (to 95.48%), but the FPS increased by 1.2 compared to the baseline model, indicating its potential in cross-scale feature fusion. In Experiment 4, the combination of WTConv and PDC-CGAD reduces the number of parameters to 13.09M and the computational cost to 42.4 GFLOPs, while maintaining the mAP50 at 96.60%, confirming the strong synergy between these methods. In Experiment 5, combining WTConv with WFU, achieves an mAP50 of 97.94% while further reducing the number of parameters and computational cost to 17.48M and 48.8 GFLOPs, respectively, reaffirming the lightweight advantage

**Table 8. Ablation experiments results.**

| Methods | WTConv | PDC-CGAD | WFU | Precision(%) | mAP@50(%) | mAP@50:95(%) | Params(M) | GFLOPs(G) | FPS(bs=1) |
|---------|--------|----------|-----|--------------|-----------|--------------|-----------|-----------|-----------|
| base    |        |          |     | 97.97 | 96.48 | 71.86 | 19.88 | 57.0 | 63.3 |
| 1       | ✓      |          |     | 98.01 | 97.32 | 71.73 | **12.83** | **40.3** | 48.6 |
| 2       |        | ✓        |     | 98.22 | 98.03 | 71.92 | 19.88 | 58.2 | 58.9 |
| 3       |        |          | ✓   | 96.54 | 95.48 | 69.64 | 23.44 | 63.2 | **64.5** |
| 4       | ✓      | ✓        |     | 94.49 | 96.60 | 70.46 | 13.09 | 42.4 | 47.9 |
| 5       | ✓      |          | ✓   | 97.09 | 97.94 | 69.74 | 17.48 | 48.8 | 51.4 |
| 6       |        | ✓        | ✓   | 97.49 | 95.02 | 68.40 | 22.85 | 60.8 | 58.9 |
| 7       | ✓      | ✓        | ✓   | **99.10** | **98.27** | **72.39** | 16.89 | 46.3 | 47.4 |

of WTConv. Finally, in Experiment 7, integrating all three methods yields the optimal performance: the mAP50 increases to 98.27%, precision reaches 99.10%, and both the parameter count and computational cost are optimized to 16.89M and 46.3 GFLOPs, respectively. The improved model has an FPS of 47.4, which meets the requirements of the real-time rail fastener target detection task. From the overall ablation experiments, WFU did not bring significant improvements in some metrics and even increased the number of parameters and computational cost. However, it significantly improved the model's inference speed, the ability to locate and distinguish targets, and the high-frequency and low-frequency responses to features by itself and in combination with other modules, proving that it did change the feature distribution. By combining it with other modules, the performance of the final model is significantly improved.

In summary, the experimental results indicate that the rational combination of WTConv, PDC-CGAD, and WFU achieves a good balance among model performance, computational efficiency, and parameter size, providing valuable insights for designing lightweight object detection models.

## Visual analysis

In the performance comparison analysis of object detection, we found significant differences among different models in the detection of railway fasteners and track components. Therefore, we selected several representative images as inference images for visual comparison. As shown in Fig 8, the model comparisons for YOLOv8, YOLOv10, YOLOv12, RT-DETRv2, RT-DETRV3, DEIM, RF-DETR, RT-DETR-R18, and the RFD-DETR model proposed in this paper are presented. Each column corresponds to a model, and each row displays the detection results of the same test image under different models, facilitating an intuitive comparison of the performance differences between models. Although these methods do not differ significantly in terms of parameter quantity and computational requirements, they all demonstrate the ability to detect key objects. However, there are still significant differences in detection accuracy, bounding box localization accuracy, and the reliability of confidence scores.

From the visualization results, it can be seen that all models perform excellently in detecting critical railway fasteners, but there are differences in detection confidence, bounding box localization, and false negatives. YOLOv8 and YOLOv10 often have higher confidence scores in their prediction results, but they may miss some targets at the image boundaries and have lower confidence scores, resulting in significant differences in detection capabilities compared to other models. YOLOv12 performs exceptionally well in target localization and detection across all aspects, but its detection confidence for certain categories (such as missing and trackbed-stuff) still lags significantly behind the model RFD-DETR in this paper. This may also be due to sample imbalance in these categories, further highlighting the superiority and advanced nature of the RFD-DETR model in actual image inference. Similarly, in the subsequent DETR variants, the baseline model RT-DETR shows a noticeable gap compared to RT-DETRv2 and RT-DETRv3. As shown in the fifth row of the inference image comparison in the figure, although the baseline model has higher confidence scores for the "missing" category than these two models, it still fails to detect the "trackbed-stuff" category. When comparing the RF-DETR and DEIM models with the RFD-DETR model, both demonstrate better consistency in object detection. However, the proposed model demonstrates superior performance in visualized prediction results. It outperforms other models in both category recognition accuracy and confidence score estimation. Although these models have slightly higher confidence scores for certain objects, the proposed model exhibits more precise bounding

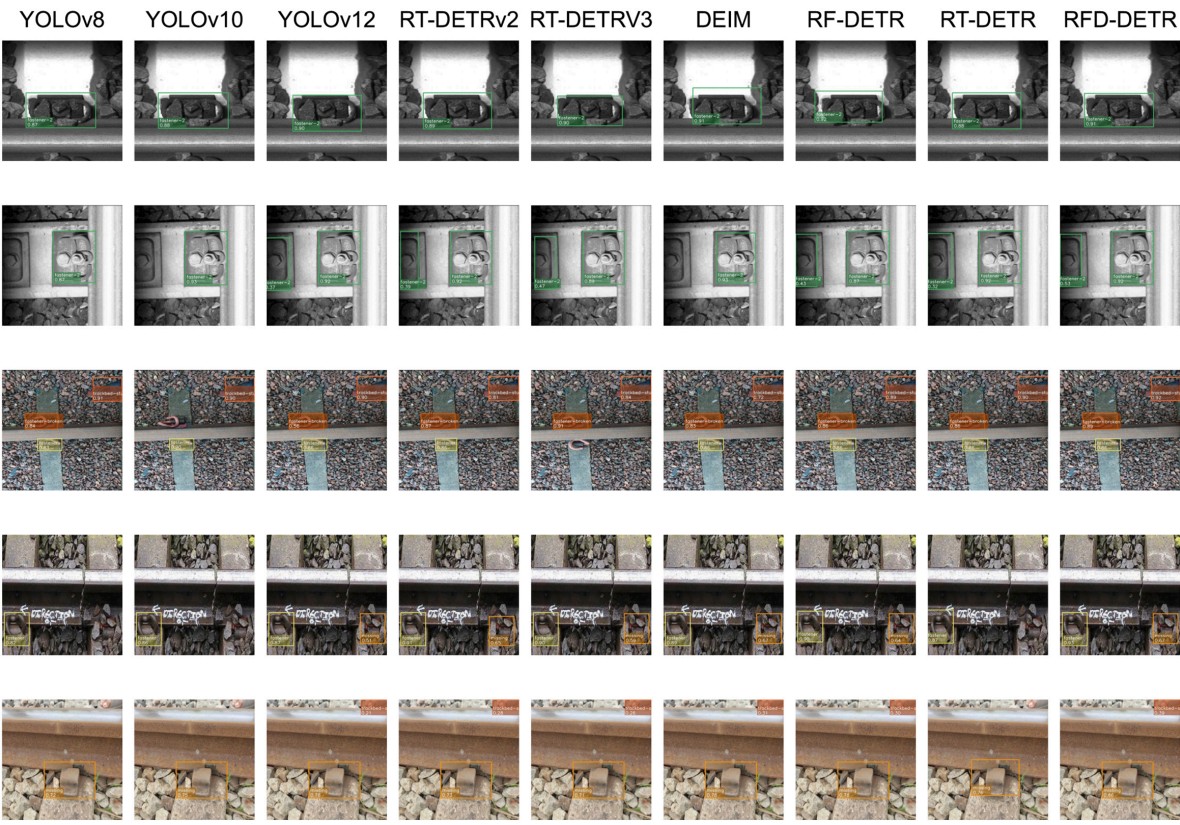

**Fig 8. Comparison of detection effects in different model.**

box localization, effectively addressing issues such as bounding box misalignment or overlap observed in competing methods.

In summary, the proposed model demonstrates robust detection performance across all object categories, exhibits stronger recognition capabilities in complex backgrounds, and possesses superior generalization potential compared to existing models. These findings highlight the advantages of RFD-DETR in railway foreign object detection, affirming its practicality and outstanding performance in real-world scenarios.

## Conclusion

This study presents RFD-DETR, a lightweight transformer-based real-time detection model specifically designed to optimize railway fastener defect detection. By integrating WTConv, CSPPDC, and WFU modules, the model enhances feature extraction, cross-scale feature fusion, and detailed information representation, significantly improving complex object detection performance while streamlining computational demands and parameter counts. These innovations position RFD-DETR as a resource-efficient solution for real-time deployment scenarios. RFD-DETR achieves a mAP@50 of 98.27%, surpassing baseline performance, while exhibiting an 18.8% reduction in computational complexity (GFLOPs) and a 14.7% decrease in parameter count. The findings demonstrate that the proposed approach achieves an optimal balance between high detection accuracy and computational efficiency.

Future applications of this research can significantly enhance intelligent railway maintenance by improving inspection efficiency, reducing manual costs, and promoting safety.

However, a primary limitation lies in the scarcity of publicly available railway fastener defect datasets, necessitating data augmentation as the principal approach for dataset expansion, which may hinder model generalization. In future work, we plan to explore the use of 3D modeling techniques to generate synthetic railway fastener images. By constructing realistic 3D models and simulating various defect types, lighting conditions, and backgrounds, we aim to create a more diverse and balanced dataset to further improve the model's generalization and robustness. Additionally, we will investigate advanced model compression strategies, such as pruning, quantization, and knowledge distillation, to facilitate the deployment of RFD-DETR on edge devices with limited computational resources. These approaches may face challenges, such as maintaining detection accuracy while reducing model size, and ensuring that synthetic data closely matches real-world scenarios. Addressing these issues will be crucial for further enhancing the practical applicability and deployment efficiency of the proposed method.

## Author contributions

**Conceptualization:** Liu Yuhao.

**Funding acquisition:** Huixiang Zhou.

**Investigation:** Liu Yuhao.

**Methodology:** Liu Yuhao.

**Software:** Jian Wang.

**Supervision:** Huixiang Zhou.

**Validation:** Liu Yuhao.

**Visualization:** Liu Yuhao.

**Writing – original draft:** Liu Yuhao.

**Writing – review & editing:** Liu Yuhao.

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
