## [Decision Letter · Decision Letter 0]

12 Jun 2025

PONE-D-25-26834Railway fastener defect detection using RFD-DETR: a lightweight real-time transformer-based approachPLOS ONE

Dear Dr. Yuhao,

Thank you for submitting your manuscript to PLOS ONE. After careful consideration, we feel that it has merit but does not fully meet PLOS ONE’s publication criteria as it currently stands. Therefore, we invite you to submit a revised version of the manuscript that addresses the points raised during the review process.

We look forward to receiving your revised manuscript.

Kind regards,

Aiqing Fang

Academic Editor

PLOS ONE

“Huixiang Zhou,

National Natural Science Foundation of China(Grant No. 62162028)”

Reviewers' comments:

Reviewer's Responses to Questions

**Comments to the Author**

1. Is the manuscript technically sound, and do the data support the conclusions?

Reviewer #1: Partly

Reviewer #2: Yes

2. Has the statistical analysis been performed appropriately and rigorously? 

Reviewer #1: No

Reviewer #2: Yes

3. Have the authors made all data underlying the findings in their manuscript fully available?

Reviewer #1: No

Reviewer #2: No

4. Is the manuscript presented in an intelligible fashion and written in standard English?

Reviewer #1: Yes

Reviewer #2: Yes

5. Review Comments to the Author

Reviewer #1: The manuscript proposes RFD-DETR, a lightweight real-time Transformer model for railway fastener defect detection, based on an augmented railway fastener dataset from the Roboflow platform. By introducing a wavelet transform convolution module, a cross-scale feature fusion module, and a wavelet feature upgrading module, the model significantly improves detection accuracy and computational efficiency. The article is well-structured, methodologically innovative, and presents convincing experimental results, offering substantial application value for intelligent railway maintenance. However, shortcomings exist in the data availability statement, robustness analysis under extreme scenarios, and comparison with the latest research. I recommend acceptance with revisions.

The paper introduces RFD-DETR, a lightweight Transformer model optimized from RT-DETR for real-time railway fastener defect detection. The model enhances performance through three innovative modules: the wavelet transform convolution module improves feature extraction via multi-scale wavelet decomposition while reducing computational load; the cross-scale feature fusion module combines cross-stage partial parallel dilated convolution and channel-gated attention downsampling to optimize multi-scale feature representation; and the wavelet feature upgrading module decomposes and reconstructs high- and low-frequency features using Haar wavelet transform to enhance feature fusion. Experiments on an augmented dataset of 2234 images, covering six fastener states, demonstrate that RFD-DETR achieves a mean average precision of 98.27% at an intersection-over-union threshold of 0.5, outperforming baseline models, with an 18.8% reduction in computational complexity and a 14.7% decrease in parameter count. The study suggests the model provides an efficient solution for railway maintenance and recommends future improvements through 3D modeling and edge computing to enhance generalization and deployment efficiency.

The article states that the dataset is sourced from the Roboflow platform at https://universe.roboflow.com/objectdetectiondeeplearning/railwaytrack_fastener_defcts1 but does not clarify whether all data are publicly available or provide details on access restrictions or contact information. I recommend supplementing a complete data availability statement to specify data access methods or limitations.

Experimental results show lower mean average precision for the “missing” and “trackbed-stuff” categories, at 60.46% and 60.03% respectively, possibly due to sample complexity or imbalanced data distribution. The paper does not deeply analyze the model’s robustness under extreme weather, severe occlusion, or complex backgrounds. I suggest adding experiments or discussions targeting these scenarios to validate the model’s practical applicability.

The literature review covers YOLO series and Transformer models but includes few references to 2024-2025 studies, such as recent advances in lightweight Transformers or railway defect detection. I recommend incorporating comparisons with the latest research to highlight RFD-DETR’s novelty.

The article thoroughly describes the technical implementation of the wavelet transform convolution module, cross-scale feature fusion module, and wavelet feature upgrading module but does not adequately explain how these modules specifically enhance feature representation for fastener defect detection. I suggest adding visualization analysis or theoretical derivations to improve model interpretability.

The experimental setup describes hardware and hyperparameter settings but omits details on the implementation of data augmentation techniques, such as the proportions or parameters for flips, scaling, and rotations. I recommend including a detailed description of data augmentation in the methods section.

The paper proposes optimizing the model through 3D modeling and edge computing but does not specify implementation pathways or potential challenges. I recommend providing more specific future research directions, such as strategies for designing 3D synthetic data generation or model compression techniques for edge devices.

The manuscript offers an efficient lightweight solution for railway fastener defect detection through the RFD-DETR model, with experimental results demonstrating significant advantages in accuracy and efficiency, holding substantial importance for intelligent railway maintenance. However, improvements are needed in the data availability statement, extreme scenario analysis, comparison with recent studies, and model interpretability. I recommend acceptance with revisions, focusing on addressing major issues, such as clarifying data availability details and conducting experiments in extreme scenarios, and refining minor issues, such as terminology consistency and figure explanations. With these revisions, the article will make a significant contribution to the field of railway defect detection.

Reviewer #2: This manuscript proposes a framework for Railway fasteners detection based on lightweight ResNet-18. The experiment verifies the effectiveness of the method, however, I have the following concerns:

1. The authors claim that their method effectively reduces computational complexity and enhances feature extraction capabilities. However, the authors only compared the FLOPS and parameter counts of different models, and the wavelet transform process does not seem to be considered, which is not a very fair comparison.

2. The authors mentioned real-time several times in the title and in the manuscript, however, there is not a single experiment in the manuscript that tests real-time performance.

3. The author's comparison method is limited to the YOLO series, which is not enough. It is recommended that the author add more than ten latest lightweight comparison methods published in top journals or conferences in the past three years.

4. Wavelet transform plays a very important role in the manuscript, however, wavelet transform experiments are missing, and the effects of different wavelet basis functions and wavelet transform parameters on the results have not been effectively evaluated.

5. Some of the technical details (e.g. formula derivation, module structure description) in the paper are lengthy, and the descriptions of the WTConv, CSPPDC and WFU modules take up a lot of space, so the innovations of each module are highlighted in the form of bold or paragraph headings

6. There are inconsistencies in the use of terminology in the paper, e.g. “necking network” should be standardized as “neck module” and “WFU” is interpreted differently in different paragraphs. WFU” is interpreted differently in different paragraphs.

6. PLOS authors have the option to publish the peer review history of their article (what does this mean?). If published, this will include your full peer review and any attached files.

Reviewer #1: No

Reviewer #2: No

---

## [Author Response · Author response to Decision Letter 1]

16 Jul 2025

Railway fastener defect detection using RFD-DETR: a lightweight real-time transformer-based approach [PONE-D-25-26834] Response to Reviewers

Firstly, we are very grateful to the Editor and Reviewers for giving us such a valuable opportunity to revise our manuscript! We appreciate the time and effort that the Editor and the Reviewers for the insightful and constructive feedback. We are convinced that the Reviewers' comments have greatly improved our manuscript and brought us a lot of inspiration.

Based on the Reviewers' comments, we have corrected the errors in the content of the manuscript, and have made extensive alterations to the structure, format, and experimental analyses of our manuscript. In response, we have addressed all of the Reviewers technical concerns with a substantial amount of new data, supported by additional detailed explanations in our revised manuscript.

Below are our point-to-point responses to the Reviewers' valuable comments, including the exact location where the change can be found in the revised manuscript. We use black bold font for the Reviewers' comments, normal black font for our responses, black italics for the content in the original manuscript, and red italics for the changes in the revised manuscript.

To Editor

Thank you for your reminder. We have carefully followed the official PLOS ONE LaTeX template https://journals.plos.org/plosone/s/latex to prepare our manuscript. The formatting, structure, and file naming fully comply with the journal’s requirements. If there are any further formatting issues, we will be happy to make additional adjustments as needed.

Thank you for your reminder. In accordance with PLOS ONE’s code sharing policy, we have made all author-generated code publicly available in a GitHub repository. The repository link and relevant information have been provided in the “Data Availability Statement” and the “Additional Information” section of the submission system. The code is shared without restriction and is intended to facilitate reproducibility and reuse, fully complying with the journal’s guidelines.

“Huixiang Zhou,

National Natural Science Foundation of China(Grant No. 62162028)”

Thank you for your guidance. We have amended the Role of Funder statement as required. Specifically, the funder (Huixiang Zhou) provided financial support for this study and contributed to project supervision. He was not involved in data collection and analysis or preparation of the original draft. The specific contributions of each author are detailed in the Author Contributions section.We have included this amended statement in the Cover Letter for your reference.

Thank you for your attention to data availability. There are no legal or ethical restrictions on sharing the data used in this study. The dataset is publicly available from the Roboflow platform at the following URL:

https://universe.roboflow.com/sinclair-9xwku/railway-track_fastener_defcts1-gfyj6 The dataset is licensed under CC BY 4.0, and all data necessary to replicate our study findings can be accessed and downloaded from this repository. We have also cited the dataset in the manuscript and provided the relevant information in the Data Availability Statement and Additional Information section of the submission system.

To Reviewer #1

1. The article states that the dataset is sourced from the Roboflow platform at https://universe.roboflow.com/objectdetectiondeeplearning/railwaytrack_fastener_defcts1 but does not clarify whether all data are publicly available or provide details on access restrictions or contact information. I recommend supplementing a complete data availability statement to specify data access methods or limitations.

Thank you for your suggestion. In the revised manuscript, we have clarified the data availability statement in the "Dataset Description" section. The dataset used in this study is publicly available from the Roboflow platform at https://universe.roboflow.com/sinclair-9xwku/railway-track_fastener_defcts1-gfyj6 and is licensed under CC BY 4.0. We have also cited the dataset as a reference in the manuscript. Please note that the dataset is provided in an augmented form; the original, unaugmented images are no longer available, and the specific augmentation parameters (such as the probability and range of flips, scaling, rotations, etc.) are not documented. Therefore, all training and evaluation in this study were conducted using the provided augmented dataset, and the precise details of the augmentation process cannot be fully reported.

Location:

1. In original manuscript:

Section “Experiment and Results”, SubSection "Dataset Description", page 9, lines 351–360.

Modified in revised manuscript:

Section “Experiment and Results”, SubSection "Dataset Description", page 10, lines 383–395.

Section "References" 36, page 22,.

2. Experimental results show lower mean average precision for the “missing” and “trackbed-stuff” categories, at 60.46% and 60.03% respectively, possibly due to sample complexity or imbalanced data distribution. The paper does not deeply analyze the model’s robustness under extreme weather, severe occlusion, or complex backgrounds. I suggest adding experiments or discussions targeting these scenarios to validate the model’s practical applicability.

Thank you for your insightful comment. In the revised manuscript, we have added a new subsection “Verification of Generalizability for Complex Scenarios” in the “Experiment and Results” section. We conducted additional experiments to evaluate the robustness of the model under various extreme weather conditions (fog, rain, snow) and complex backgrounds. The results demonstrate that the improved model maintains strong detection performance and robustness in these challenging scenarios.

Location:

1. Added in revised manuscript:

Section “Experiment and Results”, SubSection "Verification of Generalizability for Complex Scenarios", page 10.

3. The literature review covers YOLO series and Transformer models but includes few references to 2024-2025 studies, such as recent advances in lightweight Transformers or railway defect detection. I recommend incorporating comparisons with the latest research to highlight RFD-DETR’s novelty.

Thank you for your valuable suggestion. In the revised manuscript, we have updated the Introduction sections to include and discuss several recent studies from 2024 and 2025, focusing on the latest advances in lightweight Transformer models and railway defect detection. In addition, we have incorporated experimental comparisons with these state-of-the-art methods (such as YOLOv10, YOLOv11, YOLOv12, RT-DETRv2, RT-DETRv3, DEIM, RF-DETR, etc.) in both the “Comparisons with Different Detection Models” subsection and the “Visual Analysis” subsection. The visual analysis provides an intuitive comparison of detection results among these models, further highlighting the novelty and advantages of RFD-DETR.

Location:

1. Added in revised manuscript:

Section “Introduction”, page 3-4, lines 72–77, lines 90-91, lines 98-111.

Section "References", page 20-22,.

2. In original manuscript:

Section “Experiment and Results”, SubSection “Comparisons with Different Detection Models”, page 11, lines 404–418.

Section “Experiment and Results”, SubSection “Visual Analysis”, page 13, lines 462-487.

Modified in revised manuscript:

Section “Experiment and Results”, SubSection “Comparisons with Different Detection Models”, page 12-14, lines 477–529, Table 4-5.

Section “Experiment and Results”, SubSection “Visual Analysis”, page 18, lines 671-711.

4. The article thoroughly describes the technical implementation of the wavelet transform convolution module, cross-scale feature fusion module, and wavelet feature upgrading module but does not adequately explain how these modules specifically enhance feature representation for fastener defect detection. I suggest adding visualization analysis or theoretical derivations to improve model interpretability.

Thank you for your constructive suggestions. In the revised version, we have added visualization analysis and further theoretical explanations to clarify how the wavelet transform convolution module, cross-scale feature fusion module, and wavelet feature enhancement module improve the feature representation capabilities for fastener defect detection. Specifically, we have added internal ablation experiments for the wavelet convolution module, feature map visualization, and heatmap analysis (e.g., Grad-CAM, Grad-CAM++, XGrad-CAM) to demonstrate the model's improved focus on relevant regions and interpretability in defect detection tasks.

Location:

1. Added in revised manuscript:

Section “Experiment and Results”, SubSection “Main Result”, page 15-17.

5. The experimental setup describes hardware and hyperparameter settings but omits details on the implementation of data augmentation techniques, such as the proportions or parameters for flips, scaling, and rotations. I recommend including a detailed description of data augmentation in the methods section.

Thank you for your helpful suggestion. In the revised manuscript, we have clarified the data augmentation process in the “Dataset Description” section. Specifically, we state that the dataset used in this study was directly obtained from the Roboflow platform in an augmented form. The original, unaugmented images are no longer available, and the specific augmentation parameters (such as the probability and range of flips, scaling, rotations, etc.) are not documented in the dataset description. Therefore, all training and evaluation in this study were conducted using the provided augmented dataset, and the precise details of the augmentation process cannot be fully reported.

Location:

1. In original manuscript:

Section “Experiment and Results”, SubSection "Dataset Description", page 9, lines 351–360.

Modified in revised manuscript:

Section “Experiment and Results”, SubSection "Dataset Description", page 10, lines 383–395.

Section "References" 36, page 22,.

6. The paper proposes optimizing the model through 3D modeling and edge computing but does not specify implementation pathways or potential challenges. I recommend providing more specific future research directions, such as strategies for designing 3D synthetic data generation or model compression techniques for edge devices.

Thank you for your valuable suggestion. In the revised manuscript, we have expanded the “Conclusion” section to provide more specific future research directions. We now discuss the use of 3D modeling techniques to generate synthetic railway fastener images by constructing realistic 3D models and simulating various defect types, lighting conditions, and backgrounds, which can help create a more diverse and balanced dataset. Additionally, we elaborate on advanced model compression strategies, such as pruning, quantization, and knowledge distillation, to facilitate the deployment of RFD-DETR on edge devices with limited computational resources. We also mention the potential challenges, such as maintaining detection accuracy while reducing model size and ensuring the synthetic data closely matches real-world scenarios.

Location:

1. In original manuscript:

Section “Conclusion”, page 13.

Modified in revised manuscript:

Section “Conclusion”, page 19, lines 728–739.

To Reviewer #2

1. The authors claim that their method effectively reduces computational complexity and enhances feature extraction capabilities. However, the authors only compared the FLOPS and parameter counts of different models, and the wavelet transform process does not seem to be considered, which is not a very fair comparison.

Thank you for your insightful comment. In the revised manuscript, we have carefully considered the computational complexity and inference speed introduced by the wavelet transform process, especially the WFU module. In the “Ablation Study” subsection, we present detailed experiments where the WFU module is used independently. The results show that, although the WFU module slightly increases the parameter count, it significantly improves the inference speed (FPS), demonstrating its efficiency in practical deployment. Furthermore, in the “Main Result” subsection, we provide heatmap visualizations to illustrate the advantages of the WFU module in enhancing feature representation and model interpretability. These results confirm that the wavelet transform process not only maintains computational efficiency but also improves detection performance.

Location:

1. Added in revised manuscript:

Section “Experiment and Results”, SubSection “Main Result”, page 16-17, lines 590-619.

1. In original manuscript:

Section “Experiment and Results”, SubSection “Ablation Study”. page 12, lines 446-448.

Modified in revised manuscript:

Section “Experiment and Results”, SubSection “Ablation Study”, page 17, lines 647–650.

Section “Experiment and Results”, SubSection “Ablation Study”, page 17-18, lines 659–666.

2. The authors mentioned real-time several times in the title and in the manuscript, however, there is not a single experiment in the manuscript that tests real-time performance.

Thank you for your valuable comment. In the revised manuscript, we have added the frames per second (FPS) metric to the experimental results to directly evaluate the real-ti

---

## [Decision Letter · Decision Letter 1]

8 Aug 2025

PONE-D-25-26834R1Railway fastener defect detection using RFD-DETR: a lightweight real-time transformer-based approachPLOS ONE

Dear Dr. Yuhao,

Thank you for submitting your manuscript to PLOS ONE. After careful consideration, we feel that it has merit but does not fully meet PLOS ONE’s publication criteria as it currently stands. Therefore, we invite you to submit a revised version of the manuscript that addresses the points raised during the review process.

We look forward to receiving your revised manuscript.

Kind regards,

Ardashir Mohammadzadeh, Phd

Academic Editor

PLOS ONE

Journal Requirements:

Reviewer's Responses to Questions

**Comments to the Author**

1. If the authors have adequately addressed your comments raised in a previous round of review and you feel that this manuscript is now acceptable for publication, you may indicate that here to bypass the “Comments to the Author” section, enter your conflict of interest statement in the “Confidential to Editor” section, and submit your "Accept" recommendation.

Reviewer #1: All comments have been addressed

Reviewer #2: All comments have been addressed

2. Is the manuscript technically sound, and do the data support the conclusions?

Reviewer #1: Yes

Reviewer #2: Yes

3. Has the statistical analysis been performed appropriately and rigorously? 

Reviewer #1: Yes

Reviewer #2: Yes

4. Have the authors made all data underlying the findings in their manuscript fully available?

Reviewer #1: Yes

Reviewer #2: Yes

5. Is the manuscript presented in an intelligible fashion and written in standard English?

Reviewer #1: Yes

Reviewer #2: Yes

6. Review Comments to the Author

Reviewer #1: The article proposes a railway fastener defect detection method based on a lightweight real-time Transformer model (RFD-DETR), with the following key contributions and highlights: It innovatively integrates high-frequency details and low-frequency structural information through the introduction of the Wavelet Transform Convolution module (WTConv), Cross-Scale Parallel Dilated Convolution module (CSPPDC), and Wavelet Feature Upgrading module (WFU), significantly improving detection accuracy and efficiency. Experimental results show that the model achieves a mean Average Precision (mAP) of 98.27% at an IoU threshold of 0.5, while reducing computational cost by 18.8%, demonstrating high performance. The addition of Frames Per Second (FPS) metrics validates its real-time detection capability, suitable for resource-constrained environments. The model exhibits strong robustness in complex scenarios such as occlusion and lighting variations, providing a practical solution for railway maintenance. It utilizes a publicly available dataset from Roboflow (CC BY 4.0 license) and shares code, enhancing research transparency and reproducibility. Visualization analyses using Grad-CAM, Grad-CAM++, and XGrad-CAM significantly improve the model’s focus on defect regions and interpretability. However, several areas require revision:

The article mentions the use of an augmented dataset but lacks detailed descriptions of specific data augmentation methods, such as the proportions or parameters for flipping, scaling, or rotation, which limits the understanding and reproducibility of the data preprocessing process.

Despite emphasizing "real-time" performance in the title and text, the original manuscript lacks experiments directly validating real-time performance, such as FPS metrics, with supplementation only in the revised version, indicating incomplete experimental design.

The initial draft includes limited references to 2024-2025 studies on lightweight Transformers and railway defect detection, failing to fully demonstrate RFD-DETR’s novelty compared to the latest methods. It is recommended to cite: "Optimizing Insulator Defect Detection with Improved DETR Models".

Although wavelet transform is central to the model, the initial draft lacks systematic experiments on different wavelet basis functions and decomposition levels, limiting a comprehensive evaluation of wavelet transform effects.

While WTConv, CSPPDC, and WFU modules are introduced, the article lacks in-depth theoretical analysis or derivation of how these modules specifically enhance feature representation, reducing the interpretability of the model design.

We hope the authors can address these issues in their revisions.

Reviewer #2: The authors have addressed all my concerns. I have no more questions. I suggest this manuscript to be accepted.

7. PLOS authors have the option to publish the peer review history of their article (what does this mean?). If published, this will include your full peer review and any attached files.

Reviewer #1: No

Reviewer #2: No

---

## [Author Response · Author response to Decision Letter 2]

10 Aug 2025

Railway fastener defect detection using RFD-DETR: a lightweight real-time transformer-based approach [PONE-D-25-26834]

Response to Reviewers

Dear Editor and Reviewers,

We sincerely thank you for providing us with this valuable opportunity to revise our manuscript. We greatly appreciate the time and effort you have dedicated to reviewing our work and providing insightful and constructive feedback. The reviewers' comments have significantly improved our manuscript and provided us with valuable insights for future research.

Based on the reviewers' comments, we have made comprehensive revisions to address all technical concerns raised. Our revised manuscript includes substantial detailed explanations to support our findings.

Below are our point-by-point responses to each comment, with specific references to the locations of changes in our revised manuscript.

To Editor - Journal Requirements:

We have carefully reviewed our reference list to ensure completeness and correctness. After thorough verification, we confirm that:

1. All cited references are current and relevant to our research

2. No retracted papers are included in our reference list

3. All references are properly formatted according to PLOS ONE guidelines

We have adjusted the order of references cited in the main text to ensure they appear in descending chronological order, as recommended by the journal. This improves the manuscript's readability and follows standard academic citation practices.

To Reviewer #1

Thank you for your continued valuable feedback and for acknowledging that all major comments have been addressed. We appreciate your recognition of our model's contributions and the improvements made in the revised manuscript. Below are our responses to your additional suggestions:

1. The article mentions the use of an augmented dataset but lacks detailed descriptions of specific data augmentation methods, such as the proportions or parameters for flipping, scaling, or rotation, which limits the understanding and reproducibility of the data preprocessing process.

As explained in our Dataset Description section (lines 383-395), the dataset was obtained from Roboflow in an augmented form, and the original augmentation parameters are not documented by the dataset provider. We have transparently reported this limitation in the manuscript.

2. Despite emphasizing "real-time" performance in the title and text, the original manuscript lacks experiments directly validating real-time performance, such as FPS metrics, with supplementation only in the revised version, indicating incomplete experimental design.

Thank you for this important observation. You are absolutely correct that real-time performance validation is essential for a model emphasizing real-time capabilities. In the revised manuscript, we have comprehensively addressed this by incorporating Frames Per Second (FPS) metrics throughout our experimental evaluation to directly validate the real-time performance claims.

Specific additions in the revised manuscript:

Table 5 (Efficiency comparison): FPS metrics for all compared models, demonstrating that RFD-DETR achieves 47.4 FPS, which meets real-time detection requirements.

Table 8 (Ablation study): FPS evaluation for each module combination, showing the impact of different components on inference speed.

Text analysis: Discussion of real-time performance trade-offs and validation of the model's suitability for real-time railway fastener detection scenarios.

The FPS results confirm that our proposed RFD-DETR maintains competitive inference speed while achieving superior detection accuracy, thus validating the "real-time" claims in our title and methodology.

3. The initial draft includes limited references to 2024-2025 studies on lightweight Transformers and railway defect detection, failing to fully demonstrate RFD-DETR’s novelty compared to the latest methods. It is recommended to cite: "Optimizing Insulator Defect Detection with Improved DETR Models".

We appreciate the reviewers' suggestion to cite the latest research findings. In the revised manuscript, we have included the recommended reference “Optimizing Insulator Defect Detection via an Improved DETR Model” in the Introduction section (lines 114–121). This citation serves to demonstrate how the innovative RFD-DETR method compares to and surpasses the latest lightweight transformer-based defect detection methods. The addition of this contemporary reference reinforces the positioning of this manuscript within the current research field and better highlights the novelty of the method we propose.

4. Although wavelet transform is central to the model, the initial draft lacks systematic experiments on different wavelet basis functions and decomposition levels, limiting a comprehensive evaluation of wavelet transform effects.

We acknowledge the reviewer's observation that wavelet transform is indeed central to our RFD-DETR model. In the revised manuscript, we have addressed this limitation by adding comprehensive ablation experiments on different wavelet basis functions and various decomposition levels (1-3 levels) in the "Main Result" subsection. These systematic experiments provide quantitative evidence of how different wavelet configurations affect the model's performance, offering insights into the optimal wavelet parameters for railway fastener defect detection. This enhancement significantly improves the comprehensiveness of our wavelet transform evaluation and strengthens the experimental foundation of our proposed method.

5. While WTConv, CSPPDC, and WFU modules are introduced, the article lacks in-depth theoretical analysis or derivation of how these modules specifically enhance feature representation, reducing the interpretability of the model design.

We appreciate the reviewers' feedback on the theoretical depth of the modules we proposed. In the revised version, we have significantly enhanced the theoretical analysis of the WTConv, CSPPDC, and WFU modules in the “Main Result” section. Specifically, we have added comprehensive visualization experiments and theoretical explanations to demonstrate how each module contributes to the enhancement of feature representation. For WTConv, we provide a detailed analysis through ablation comparisons using different wavelet basis functions and different numbers of wavelet decomposition layers, illustrating how the wavelet transform captures multi-scale frequency information. For CSPPDC, we demonstrate its cross-scale feature fusion capabilities through comparative visualization analysis. For WFU, we use heatmaps and Grad-CAM visualizations to detail the mechanism of wavelet feature enhancement. These experimental analyses, combined with theoretical explanations, significantly enhance the interpretability of the model design and provide readers with concrete evidence of how our modules enhance feature representation.

To Reviewer #2

We sincerely thank Reviewer #2 for their thorough review and for confirming that all previous comments have been adequately addressed. We appreciate their recommendation for acceptance and are grateful for their constructive feedback throughout the review process.

We believe that the revised manuscript now addresses all the concerns raised by the reviewers and meets the publication standards of PLOS ONE. We have provided comprehensive responses to each comment and made substantial improvements to enhance the manuscript's quality, clarity, and scientific rigor.

We sincerely thank the Editor and Reviewers for their valuable time and constructive feedback, which have significantly improved our manuscript.

Sincerely,

The Authors

---

## [Decision Letter · Decision Letter 2]

17 Aug 2025

Railway fastener defect detection using RFD-DETR: a lightweight real-time transformer-based approach

PONE-D-25-26834R2

Dear Dr. Yuhao,

We’re pleased to inform you that your manuscript has been judged scientifically suitable for publication and will be formally accepted for publication once it meets all outstanding technical requirements.

Kind regards,

Ardashir Mohammadzadeh, Phd

Academic Editor

PLOS ONE

Additional Editor Comments (optional):

Reviewers' comments:

Reviewer's Responses to Questions

**Comments to the Author**

1. If the authors have adequately addressed your comments raised in a previous round of review and you feel that this manuscript is now acceptable for publication, you may indicate that here to bypass the “Comments to the Author” section, enter your conflict of interest statement in the “Confidential to Editor” section, and submit your "Accept" recommendation.

Reviewer #1: All comments have been addressed

2. Is the manuscript technically sound, and do the data support the conclusions?

Reviewer #1: Yes

3. Has the statistical analysis been performed appropriately and rigorously? 

Reviewer #1: Yes

4. Have the authors made all data underlying the findings in their manuscript fully available?

Reviewer #1: Yes

5. Is the manuscript presented in an intelligible fashion and written in standard English?

Reviewer #1: Yes

6. Review Comments to the Author

Reviewer #1: This version of the revised manuscript is better. The author has answered all of my questions. The article has reached publication standard. Recommended

7. PLOS authors have the option to publish the peer review history of their article (what does this mean?). If published, this will include your full peer review and any attached files.

Reviewer #1: No

---

## [Editor Report · Acceptance letter]

PONE-D-25-26834R2

PLOS ONE

Dear Dr. Yuhao,

I'm pleased to inform you that your manuscript has been deemed suitable for publication in PLOS ONE. Congratulations! Your manuscript is now being handed over to our production team.

Kind regards,

on behalf of

Dr. Ardashir Mohammadzadeh

Academic Editor

PLOS ONE